# Volumetric Optimal Transportation by Fast Fourier Transform

**Na Lei** [*]
Dalian University of Technology
nalei@dlut.edu.cn

**Dongsheng An** [*]
Stony Brook University
doan@cs.stonybrook.edu

**Min Zhang**
Zhejiang University
min_zhang@zju.edu.cn

**Xiaoyin Xu**
Harvard Medical School
xxu@bwh.harvard.edu

**Xianfeng Gu**
Stony Brook University
gu@cs.stonybrook.edu

## Abstract

The optimal transportation map finds the most economical way to transport one probability measure to another, and it has been applied in a broad range of applications in machine learning and computer vision. By the Brenier theory, computing the optimal transport map is equivalent to solving a Monge-Ampère equation, which is highly non-linear. Therefore, the computation of optimal transportation maps is intrinsically challenging. In this work, we propose a novel and powerful method, the FFT-OT (fast Fourier transform-optimal transport), to compute the 3-dimensional OT problems. The method is based on several key ideas: first, the Monge-Ampère equation is linearized to a sequence of linear elliptic PDEs with spacial and temporal variant coefficients; second, the obliqueness property of optimal transportation maps is reformulated as a Neumann boundary condition; and third, the variant coefficient elliptic PDEs are approximated by constant coefficient elliptic PDEs and solved by FFT on GPUs. We also prove that the algorithm converges linearly. Experimental results show that the FFT-OT algorithm is more than a hundred times faster than the conventional methods based on the convex geometry. Furthermore, the method can be directly applied for sampling from complex 3D density functions in machine learning and magnifying the volumetric data in medical imaging.

## 1 Introduction

Optimal transportation (OT) transports one probability measure to another in the most economical way, and it plays a fundamental role in areas like machine learning Courty et al. (2017); Altschuler et al. (2019), computer vision Arjovsky et al. (2017); Tolstikhin et al. (2018); An et al. (2020), and computer graphics Solomon et al. (2015); Nader & Guennebaud (2018). Given a Riemannian manifold $X$, all the probability distributions on $X$ form an infinite dimensional space $\mathcal{P}(X)$. Given any two distributions $\mu, \nu \in \mathcal{P}(X)$, the optimal transportation map defines a distance between them, and the McCann interpolation McCann (1997) defines the geodesic connecting them. Hence optimal transportation equips $\mathcal{P}(X)$ with a Riemannian metric and defines its covariant differentiation, which provides a variational calculus framework for optimization in it.

As the optimal transportation problem is highly non-linear, it is quite challenging to compute the OT maps. Recently, researchers have developed many algorithms. The geometric variational approach Aurenhammer et al. (1998); Gu et al. (2016); Levy (2015) based on the Brenier theorem Brenier (1991) is capable of achieving high accuracy for low dimensional problems, but it requires complicated geometric data structure and the storage complexity grows exponentially as the dimension increases. The Sinkhorn method Cuturi (2013) based on the Kantorovich theorem adds an entropic regularizer to the primal problem and can handle high dimensional tasks, but it suffers from the intrinsic approximation error.

---

[*] indicates equal contribution

We propose a novel method to tackle this challenging problem through Fast Fourier Transformation (FFT). According to the Brenier theorem Brenier (1991), under the quadratic distance cost, the optimal transportation map is the gradient of the Brenier potential, which satisfies the Monge-Ampère equation. With the continuity method Delanoë (1991), the Monge-Ampère equation can be linearized as a sequence of elliptic partial differential equations (PDEs) with spacial and temporal variant coefficients. By iteratively solving the linearized Monge-Ampère equations, we can obtain the OT map. Specifically, we propose to approximate the linearized Monge-Ampère equation by constant coefficient elliptic PDEs and solve them using the FFT on GPUs.

Our proposed FFT-OT method has many merits: (i) it is generalizable for arbitrary dimension; (ii) it has a linear convergence rate, namely the approximation error decays exponentially fast; (iii) in each iteration, the computational complexity of FFT is $O(n \log n)$, thus our algorithm can solve large scale OT problems; and (iv) it is highly parallelable and can be efficiently implemented on GPUs. We demonstrate the efficiency of the FFT-OT algorithm by solving the volumetric OT problems for machine learning and medical imaging applications including sampling from given 3D density functions and volumetric magnifier. The algorithm also has its own limitations: (i) although it can be generalized to any dimensions, the storage complexity increase exponentially with respect to the dimension, so its power is limited by the memory size of the GPUs; (ii) Since the algorithm uses FFT, the current version of the method only works well for continuous density functions. (iii) In this work, we mainly focus on the computation of the OT map from the uniform distribution to another arbitrary continuous distribution. To extend the method to find the OT map between any two continuous measures, we can compute two OT maps from the uniform distribution to the both continuous measures, then combine them together. The combination will give a reasonable approximation of the OT map Nader & Guennebaud (2018).

Though Lei and Gu Lei & Gu (2021) also uses FFT to solve the 2-dimensional OT problem, our method differs their works in the following two aspects: (i) Lei and Gu's method uses the fixed point method to compute the 2D OT problems, ours is based on the linearization of the Monge-Ampère operator to solve the 3D OT problems, these are two different methodologies in PDE theory; (ii) In our paper, we also provide the theoretical convergence analysis of the proposed method. For more detailed analysis and related work, please refer to the Appendix A.

## 2 OPTIMAL TRANSPORTATION THEORY

In this section, we review the fundamental concepts and theorems of the OT problem and the Monge-Amperè equation, more details can be found in Villani (2008).

**Optimal Transportation Map and the Monge-Ampère equation**   Suppose the source domain $\Omega$ is an open set in $\mathbb{R}^d$ with the probability measure $\mu$, the target domain $\Sigma$ is with the probability measure $\nu$. Both $\mu$ and $\nu$ have density functions $d\mu(x) = f(x)dx$ and $d\nu(y) = g(y)dy$, respectively, with the equal total mass: $\int_\Omega f(x)dx = \int_\Sigma g(y)dy$, which is called the *balance condition*.

Suppose $T : \Omega \to \Sigma$ is a measurable map. The mapping $T$ is called *measure preserving* and denoted as $T_\# \mu = \nu$ if the following relation

$$\mu(T^{-1}(A)) = \nu(A) \tag{1}$$

for every Borel subset $A \subset \Sigma$. A *cost function* $c : \Omega \times \Sigma \to \mathbb{R}$ measures the transportation cost for transporting the unit mass from $x \in \Omega$ to $y \in \Sigma$.

**Problem 1** (Monge). *The optimal transportation problem finds the measure preserving map with the minimal total transportation cost,*

$$\min_{T_\# \mu = \nu} \int_\Omega c(x, T(x)) f(x) dx$$

The solution to the Monge's problem is called the *optimal transport map* between $\mu$ and $\nu$. The existence, uniqueness and regularity of OT maps depend on the boundedness and the continuity of the density functions, the convexity of the supporting domains, the continuity of their boundaries, and the cost function. In our current work, we focus on the similar situation in Saumier et al. (2013),

- The cost function is quadratic Euclidean distance $c(x, y) = \|x - y\|^2 / 2$;

- The supports of the source and the target measures are the canonical cube $\Omega = [-1,1]^3$, which is uniformly convex;
- The source and the target measures $\mu, \nu$ are absolutely continuous with respect to the Lebesgue measure, their densities $f, g$ are positive and bounded away from zero;
$$0 < m < f, g < M,$$
and $f, g$ are of class $C^\alpha(\Omega)$,
- The boundary condition is second boundary condition (OT boundary condition), $T(\Omega) = \Omega$.

Then according to (Villani (2003) Theorem 14.4, Saumier et al. (2013) Theorem 2.1), the OT maps $T : \Omega \to \Omega$ exists and is unique and invertible ($\mu$ a.e), and the Brenier potential is of class $C^{2,\beta}(\Omega)$ form some $0 < \beta < \alpha$.

**Theorem 2.** *Assume that $\Omega$, $\mu, \nu$, $f$ and $g$ are defined as above. Then there exists a convex function $u : \Omega \to \mathbb{R}$, $u \in C^{2,\beta}(\Omega)$ for some $0 < \beta < \alpha$, such that $\nabla u$ pushes $\mu$ forward to $\nu$, $(\nabla u)_{\#}\mu = \nu$. Moreover, $\nabla u$ is unique and invertible ($\mu$ a.e), and its inverse $\nabla v$ satisfies $(\nabla v)_{\#}\nu = \mu$.*

We call such a convex function $u$ the *Brenier potential*, it satisfies the Monge-Ampère equation,

$$\det D^2 u(x) = \frac{f(x)}{g \circ \nabla u(x)}. \tag{2}$$

with the boundary condition $\nabla u(\Omega) = \Sigma$. Then finding the optimal transportation map is equivalent to solving the corresponding Monge-Ampère equation. In the current work, the target measure is always the Lebesgue measure, and the source density $f$ is of class $C^{2,\alpha}(\Omega)$.

**Linearized Monge-Ampère Operator**  The Monge-Ampère operator is defined as

$$\mathrm{MA}[u] = \det D^2 u,$$

which is highly non-linear. It can be linearized as following:

$$\mathrm{MA}[u + \varepsilon v] = \det(D^2 u + \varepsilon D^2 v) \approx \det D^2 u + \varepsilon \mathrm{Trace}(\mathrm{Adj}(D^2 u) \cdot D^2 v), \tag{3}$$

where $\mathrm{Adj}(A)$ is the adjoint (co-factor) matrix of $A$, $\mathrm{Adj}(A) := \det(A)A^{-T}$. Therefore the *linearized Monge-Ampère operator* is defined as

$$\mathrm{DMA}_u[v] := \mathrm{Trace}(\mathrm{Adj}(D^2 u) \cdot D^2 v) = \sum_{p,q=1}^{d} u^{pq}(x)\partial_p\partial_q v(x), \tag{4}$$

where $(u^{pq}) = \mathrm{Adj}(D^2 u)$ is the adjoint matrix of the Hessian of $u$, and $\partial_p\partial_q := \frac{\partial^2}{\partial x_p \partial x_q}$.

**Continuity Method**  For simplicity, we assume the source domain coincides with the target domain, that is $\Omega = \Sigma$, and the target density is $g(x) \equiv 1$. The Monge-Ampère equation Eqn. (2) is simplified as $\det D^2 u(x) = f(x)$. Define a flow of density as

$$\rho(x,t) = (1-t) + tf(x), \quad t \in [0,1]. \tag{5}$$

The corresponding flow of the Brenier potentials is $u(x,t) : \Omega \times [0,1] \to \mathbb{R}$,

$$\det D_x^2 u(x,t) = \rho(x,t), \quad s.t. \ \nabla_x u(x,t)(\Omega) = \Omega,$$

where $D_x^2 u(x,t)$ is the Hessian of $u(x,t)$ with respect to $x$, and $u(x,1)$ is the solution to the initial Monge-Ampère equation Eqn. (2). Take the derivative w.r.t. time $t$ on both sides of the linearized Monge-Ampère operator Eqn. (4), we obtain an elliptic PDE with the spacial and temporal variant coefficients of the unknown $v(x,t) := \dot{u}(x,t)$, namely the "velocity" of the Brenier potential,

$$\mathrm{DMA}_u[v] = \sum_{p,q=1}^{d} u^{pq}(x,t)\partial_p\partial_q v(x,t) = \frac{\partial}{\partial t}\rho(x,t) = f(x) - 1. \tag{6}$$

At time $t = 0$, the initial Brenier potential is known as $u(x,0) = \frac{1}{2}\|x\|^2$. Suppose at time $t$, we have obtained $u(x,t)$ already, then we can compute the adjoint matrix $u^{pq}(x,t)$ of the Hessian $D_x^2 u(x,t)$, and solve Eqn. (6) to get the velocity $v(x,t) = \dot{u}(x,t)$. In turn, we move forward to time $t + \delta t$, and update $u(x, t+\delta t)$ by $u(x,t) + \dot{u}(x,t)\delta t$. By repeating this procedure, eventually we reach time $t = 1$ and obtain the solution $u(x) := u(x,1)$ to the initial Monge-Ampère Eqn. (2).

**Obliqueness Boundary Condition**   Suppose the boundary of $\Omega$ is $C^1$ almost everywhere, therefore at a $C^1$ point $x \in \partial\Omega$, the outer normal $\mathbf{n}(x)$ is well defined. For almost every boundary point $x \in \partial\Omega$, the obliqueness condition is represented as

$$\langle \mathbf{n}(x), \mathbf{n}(\nabla u(x)) \rangle \geq 0. \tag{7}$$

Suppose $\Omega$ is a cuboid and has 6 faces, if a boundary point $x \in \partial\Omega$ is on a face, by the cyclic monotonicity of the map and the strict convexity of $u$ Villani (2008), its image $\nabla u(x)$ must be on the same face of $x$, namely,

$$\langle \nabla u(x) - x, \mathbf{n}(x) \rangle = 0. \tag{8}$$

We can rewrite the Brenier potential as $u(x_1, x_2, \ldots, x_d) = \frac{1}{2}\sum_{i=1}^{d} x_i^2 + v(x_1, \cdots, x_d)$, then $\nabla u(x) - x = \nabla v(x)$. By Eqn. (8), $v(x)$ satisfies the Neumann boundary condition,

$$\frac{\partial v}{\partial \mathbf{n}}(x) = 0, \quad x \in \partial\Omega. \tag{9}$$

Similarly, the velocity of the (modified) Brenier potential $v$ in Eqn. (6) also satisfies the Neumann boundary condition. The analysis about the existence and regularity of the solutions to Eqn. (6) with boundary condition Eqn. (9) can be found in the supplementary material.

## 3   COMPUTATIONAL ALGORITHM

Here we introduce the 3-dimensional FFT-OT algorithm, which can be generalized to any dimensions. We approximate the Monge-Ampère equation by a sequence of constant coefficient elliptic PDEs, and solve them by FFT on GPUs. More detailed analysis about the solution of the discretized Monge-Ampère equation, and the proofs of the lemmas and theorems are given by Appendix B.

### 3.1   CONTINUITY METHOD FOR SOLVING THE MONGE-AMPÈRE EQUATION

By using the continuity method, we can solve the Monge-Ampère equation iteratively. For simplicity, we assume the target measure is the Lebesgue's measure with $g \equiv 1$. At the $n$-th iteration, the Brenier potential is represented as $\frac{1}{2}\|x\|^2 + u_n(x)$, its Hessian matrix is $H_n(x) := \mathrm{I} + D^2 u_n(x)$, the corresponding density function is defined as the determinant of the Hessian $\rho_n = \det(H_n)$, and the velocity of the Brenier potential is $v_n(x)$. In the beginning, the Brenier potential $u_0(x)$ is zero, the Hessian is $H_0 = \mathrm{I}$ and the density is $\rho_0 = 1$. At the $n$-th step, we compute the adjoint matrix $[H_n^{pq}(x)]$ of the Hessian matrix $H_n(x)$ for any $x \in \Omega$. According to Eqn. (3), the velocity $v_n(x)$ satisfies the variant coefficient elliptic PDE induced by the linearized Monge-Ampère operator,

$$\mathrm{DMA}_{u_n}[v_n] = \sum_{p,q=0}^{2} H_n^{pq}(x)\partial_p\partial_q v_n(x) = \frac{1}{\tau}(f(x) - \rho_n(x)). \tag{10}$$

Note that the right hand side of Eqn. (6) is the difference between the initial and the target densities, whereas here it is replaced by the difference between the initial and the current densities. The step length parameter $\tau \geq 1$ can be chosen to guarantee the convergence Loeper & Rapetti (2005).

The elliptic PDE Eqn. (10) is with spatially variant coefficients. Although the traditional finite element method (FEM) can solve it using the GMRES algorithm Saad (2003), this algorithm can not be directly accelerated by GPUs. To overcome this difficulty, we approximate Eqn. (10) by a much simpler elliptic PDE with constant coefficients, which can be directly solved using the following FFT-OT algorithm pipeline Alg. 1 on GPUs in Appendix C.

At the $n$-th iteration, after obtaining the adjoint matrix $[H_n^{pq}(x)]$, $x \in \Omega$, we compute the mean adjoint matrix $[\bar{H}_n^{pq}(x)]$

$$\bar{H}_n^{pq} := \frac{\int_\Omega H_n^{pq}(x)\rho_n(x)dx}{\int_\Omega \rho_n(x)dx}, \quad p, q = 0, 1, 2 \tag{11}$$

and replace the elliptic PDE Eqn.(10) with variant coefficients by the elliptic PDE with constant coefficients,

$$\overline{\mathrm{DMA}}_{u_n}[v_n] = \sum_{p,q=0}^{2} \bar{H}_n^{pq}\partial_p\partial_q v_n(x) = \frac{1}{\tau}(f(x) - \rho_n(x)), \tag{12}$$

where $\overline{\text{DMA}}$ is called the mean linearized Monge-Ampère operator.

Then we solve the constant coefficient elliptic PDE Eqn. (12) by FFT Algorithm Alg. 2 in Appendix C. Although the original variant coefficient PDE Eqn. (10) is replaced by its constant coefficient approximation Eqn. (12), the algorithm still converges to the solution with a linear convergence rate. This replacement allows the whole algorithm to be solved by FFT on GPUs, which greatly improves the computational efficiency.

**Theorem 3** (main). *Given a domain $\Omega \subset \mathbb{R}^d$, which is a canonical cuboid $\Omega = [-1, 1]^d$, and a positive density function $f : \Omega \to \mathbb{R}$ with the balance condition $\int_\Omega f(x)dx = \int_\Omega dx$, suppose the mirror reflection extension Eqn. (14) of $f$ to the flat torus $\tilde{f} : \mathbb{T}^n \to \mathbb{R}$ is $C^\alpha$, $\alpha \in (0, 1)$, then the Monge-Ampère equation,*

$$det D^2 u(x) = f(x), \quad \nabla u(\Omega) = \Omega$$

*can be solved using the FFT-OT Algorithm Alg. 1 in Appendix C. In particular, one can choose the step length parameter $\tau$, such that there is a constant $0 < \gamma < 1$ that the approximation error satisfies*

$$\|f - \rho_{n+1}\|^2 < C\gamma^n, \tag{13}$$

*namely the algorithm has a linear convergence rate.*

### 3.2 FFT SOLVER FOR CONSTANT COEFFICIENT ELLIPTIC PDES

To solve the constant coefficient elliptic PDE Eqn. (12), we first extend the PDE to the flat torus by mirror reflection, then discretize the domain and compute the differential operators by central difference scheme. Finally the PDE is converted to algebraic equations in the frequency domain by FFT and can be efficiently solved on GPUs.

**Extension by Mirror Reflection**  Suppose $\Omega = [0, 1]^3$ and $f : \Omega \to \mathbb{R}$ are given, we extend $\Omega$ to $\tilde{\Omega} = [-1, 1]^3$ and $f$ to $\tilde{f} : \tilde{\Omega} \to \mathbb{R}$ by mirror reflection

$$\tilde{f}(x, y, z) = f(|x|, |y|, |z|), \quad \forall (x, y, z) \in \tilde{\Omega}. \tag{14}$$

By definition, $\tilde{f}$ satisfies the periodic boundary condition and can be treated as a function defined on the flat torus $\mathbb{T}^3$. $\tilde{\Omega}$ is one of the fundamental domain of $\mathbb{T}^3$. The constant coefficients $a^{p,q}$ keep unchanged. Then we solve the following constant coefficient elliptic PDE Eqn. (18) $L[\tilde{u}] = \tilde{f}$ with the periodic boundary condition. Finally, the restriction of $\tilde{u}$ on $\Omega$ gives the initial solution $u$ to $L[u] = f$ with Neumann boundary condition.

In the following, to avoid using overly complicated symbols, we use $(u, f, \Omega)$ to represent $(\tilde{u}, \tilde{f}, \tilde{\Omega})$ for simplicity.

**Tessellation**  Suppose $\Omega = [-1, 1]^3$ is the canonical cube (a fundamental domain of a flat torus), we tessellate it to the regular cells, and the centers of the cells form a grid $M \times N \times L$. The Brenier potential $u : \Omega \to \mathbb{R}$ is discretized to a tensor $u_{i,j,k}$ with $\{i, j, k\} \in \{0, \ldots, M-1\} \times \{0, \ldots, N-1\} \times \{0, \ldots, L-1\}$. The spacial step lengths are $(h_x, h_y, h_z) = (2/M, 2/N, 2/L)$. The coordinate of each sample point $(x_i, y_j, z_k)$ is $(x_i, y_j, z_k) = (-1 + h_x(i + 1/2), -1 + h_y(j + 1/2), -1 + h_z(k + 1/2))$. The periodic boundary condition is then formulated as

$$u_{i,j,k} = u_{i+\alpha M, j+\beta N, k+\gamma L}, \quad \alpha, \beta, \gamma \in \mathbb{Z}. \tag{15}$$

**Finite Difference Differential Operator**  We use the standard central differences to compute the differential operators. The first order derivative $\mathcal{D}_x$ is approximated by

$$\mathcal{D}_x u_{i,j,k} = \frac{u_{i+1,j,k} - u_{i-1,j,k}}{2h_x},$$

where the index $i + 1$ means $i + 1$ modulus $M$. The operators $\mathcal{D}_y, \mathcal{D}_z$ are defined in a similar way. The second order derivative operator $\mathcal{D}_{xx}$ and $\mathcal{D}_{xy}$ are approximated by

$$\mathcal{D}_{xx}^2 u_{i,j,k} = \frac{u_{i+1,j,k} + u_{i-1,j,k} - 2u_{i,j,k}}{h_x^2}$$

$$\mathcal{D}_{xy}^2 u_{i,j,k} = \frac{u_{i+1,j+1,k} + u_{i-1,j-1,k} - u_{i+1,j-1,k} - u_{i-1,j+1,k}}{4h_x h_y}$$

The other operators $\mathcal{D}_{yy}, \mathcal{D}_{zz}, \mathcal{D}_{yz}$ and $\mathcal{D}_{xz}$ are defined similarly.

**Discrete Fourier Transformation**    The discrete Fourier transformation (DFT) of $u_{i,j,k}$ is given by

$$\hat{u}_{m,n,l} = \sum_{i=0}^{M-1} \sum_{j=0}^{N-1} \sum_{k=0}^{L-1} u_{i,j,k} \hat{\omega}_{mnl} \tag{16}$$

$$u_{i,j,k} = \frac{1}{MNL} \sum_{m=0}^{M-1} \sum_{n=0}^{N-1} \sum_{l=0}^{L-1} \hat{u}_{m,n,l} \omega_{mnl} \tag{17}$$

where $\hat{\omega}_{mnl} = e^{-\iota \frac{2\pi mi}{M}} e^{-\iota \frac{2\pi nj}{N}} e^{-\iota \frac{2\pi lk}{L}}$, $\omega_{mnl} = e^{\iota \frac{2\pi mi}{M}} e^{\iota \frac{2\pi nj}{N}} e^{\iota \frac{2\pi lk}{L}}$ and $\iota = \sqrt{-1}$, $\{m, n, l\}$ are the indices of the frequency coefficients. By using DFT, the differential operators are converted to algebraic operators in the frequency domain.

**Lemma 4.** *Suppose the discrete function is $u_{i,j,k}$, with the discrete Fourier transformation Eqn. (16) and Eqn. (17), by using the central difference scheme, the first order differential operator is given by*

$$\mathcal{D}_x u_{i,j,k} = \frac{1}{MNL} \sum_{m=0}^{M-1} \sum_{n=0}^{N-1} \sum_{l=0}^{L-1} \hat{u}_{m,n,l} \frac{\sin \frac{2\pi m}{M}}{h_x} \omega_{mnl}$$

*the second order differential operators are represented by*

$$\mathcal{D}_{xx}^2 u_{i,j,k} = \frac{1}{MNL} \sum_{m=0}^{M-1} \sum_{n=0}^{N-1} \sum_{l=0}^{L-1} \hat{u}_{m,n,l} \frac{2\left(\cos \frac{2\pi m}{M} - 1\right)}{h_x^2} \omega_{mnl}$$

$$\mathcal{D}_{xy}^2 u_{i,j,k} = \frac{1}{MNL} \sum_{m=0}^{M-1} \sum_{n=0}^{N-1} \sum_{l=0}^{L-1} \hat{u}_{m,n,l} \frac{-\sin \frac{2\pi m}{M} \sin \frac{2\pi n}{N}}{h_x h_y} \omega_{mnl}$$

The other differential operators $\mathcal{D}_y$, $\mathcal{D}_z$, $\mathcal{D}_{yy}$, $\mathcal{D}_{zz}$, $\mathcal{D}_{yz}$ and $\mathcal{D}_{xz}$ are also represented accordingly. The detailed proofs can be found in the supplementary material.

**FFT Solver**    Suppose we want to solve an elliptic PDE with constant coefficients on $\Omega \subset \mathbb{R}^3$,

$$L[u] := \left( \sum_{p=0}^{2} \sum_{q=0}^{2} a^{p,q} \partial_p \partial_q + \sum_{r=0}^{2} b^r \partial_r + c \right) u(x) = f(x), \tag{18}$$

with the periodic boundary condition, where $a^{p,q}, b^r, c$ are constants, the matrix $(a^{p,q})$ is positive definite, namely the PDE is uniformly elliptic. By the discrete Fourier transformation $\mathcal{F}$, we convert the differential equation to an algebraic equation in the frequency domain,

$$\sum_{p=0}^{2} \sum_{q=0}^{2} a^{p,q} \mathcal{F}(\partial_p \partial_q u) + \sum_{r=0}^{2} b^r \mathcal{F}(\partial_r u) + c \mathcal{F}(u) = \mathcal{F}(f)$$

By applying Lemma 4 and defining

$$
\begin{aligned}
\lambda_{m,n,l} = &a^{0,0} \frac{2(\cos \frac{2\pi m}{M} - 1)}{h_x^2} + a^{1,1} \frac{2(\cos \frac{2\pi n}{N} - 1)}{h_y^2} \\
&+ a^{2,2} \frac{2(\cos \frac{2\pi l}{L} - 1)}{h_z^2} - (a^{0,1} + a^{1,0}) \frac{\sin \frac{2\pi m}{M} \sin \frac{2\pi n}{N}}{h_x h_y} \\
&- (a^{1,2} + a^{2,1}) \frac{\sin \frac{2\pi n}{N} \sin \frac{2\pi l}{L}}{h_y h_z} - (a^{0,2} + a^{2,0}) \frac{\sin \frac{2\pi l}{L} \sin \frac{2\pi m}{M}}{h_z h_x} \\
&+ b^0 \frac{\sin \frac{2\pi m}{M}}{h_x} + b^1 \frac{\sin \frac{2\pi n}{N}}{h_y} + b^2 \frac{\sin \frac{2\pi l}{L}}{h_z} + c
\end{aligned} \tag{19}
$$

We have the algebraic equations in frequency domain,

$$\hat{u}_{m,n,l} \lambda_{m,n,l} = \hat{f}_{m,n,l}$$

With $\hat{u}_{m,n,l}$'s, we can easily obtain $u_{i,j,k}$'s by the Inverse Discrete Fourier Transform (IDFT), which means solving the constant coefficient elliptic equation. The algorithm is described in Alg. 2 in Appendix C.

The FFT for solving the constant coefficient elliptic PDE can be efficiently computed with GPUs. Moreover, the algorithm Alg. 2 solves the constant coefficient elliptic PDEs with a *periodic boundary condition*, which can be generalized to solving the same type of PDEs with *Neumann boundary condition* by extending the PDE to the flat torus $\mathbb{T}^3$ using mirror reflection Eqn. (14).

# 4 EXPERIMENTAL RESULTS

In this section, we firstly show that the our proposed FFT-OT algorithm converges linearly and runs $100\times$ faster than the conventional convex geometry based solver Levy (2015), then demonstrate the method in two applications: 3D adaptive sampling and Volume Magnifier. All the algorithms are developed using generic C++ with CUDA Toolkit. All the experiments are conducted on a Windows laptop with Intel Core i7-7700HQ CPU with 16 GB memory and NVIDIA GeForce GTX 1060 Graphics Cards. More experiments can be found in Appendix D.

## 4.1 RUNNING TIME AND CONVERGENCE ANALYSIS

To show the performance of the proposed method, we experiment on the density functions defined by the Gaussian mixture models. To be specific, the domain is a cube $\Omega = [0,1]^3$, the 3-dimensional density function defined on $\Omega$ is set to be $f(x) = \sum_{i=1}^{30} p_i \mathcal{N}(\mu_i, \Sigma_i)$, where $\mathcal{N}(\mu_i, \Sigma_i)$ represents Gaussian distribution with mean $\mu_i$ and variance $\Sigma_i = \text{diag}(\sigma_{i0}^2, \sigma_{i1}^2, \sigma_{i2}^2)$. $\mu_i \in \mathbb{R}^3$ is uniformly sampled from $[0,1]^3$, $\sigma_{ij}$ is uniformly sampled from $[0, 0.5]$, $p_i \in \mathbb{R}$ is uniformly sampled from $[0.2, 1]$ and normalized such that $\int_\Omega f(x)dx = 1$. Thus the source distribution $\mu$ is a complicated Gaussian mixture distribution restricted on $\Omega$. Then by mirror reflection in Sec. 3.2, we obtain the complex density function which is defined on $[-1,1]^3$ and satisfies the periodic boundary condition.

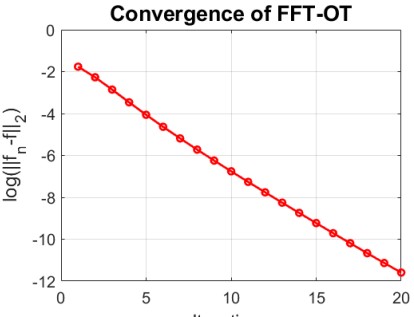

**Figure 1:** Convergence Analysis.

We directly use the FFT-OT algorithm Alg. 1 to solve the linearized Monge-Ampère equation. With the approximation error threshold $\varepsilon = 1.0 \times 10^{-6}$ and the resolution $256 \times 256 \times 256$, the running time for our FFT-OT algorithm with double precision on GPU is less than 175 seconds. The conventional convex geometry based algorithm for 3D optimal transportation Levy (2015) can neither handle such large data sets nor be implemented on GPUs. It can only compute OT map with resolution no greater than $100 \times 100 \times 100$ on our system, which takes about 2700 seconds. When handling problem with $128 \times 128 \times 128$ resolution, our FFT-OT consumes about 20.3 seconds, which is $130\times$ faster than the power diagram based method Levy (2015).

Fig. 1 shows the approximation error for the above Gaussian mixture density with respect to iterations, namely $\log \|f - \rho_n\|_2^2$. Our algorithm does converge linearly and the result is consistent with the prediction Eqn. (13) in Thm. 3. Therefore, this experiment validates the theorem.

## 4.2 3D ADAPTIVE SAMPLING

Generating random samples matching a given density function plays an essential role in the applications like Monte-Carlo integration or stippling. Efficiently obtaining high quality samples is still an on-going research topic Bauer et al. (2015); Perrier et al. (2018). And optimal transportation has been successfully applied for generating high quality 2D samples de Goes et al. (2012); Nader & Guennebaud (2018). Most of the current research focuses on generating 2D samples fitting the given density function. Here we apply the proposed 3D FFT-OT method to generate high quality 3D samples according to the given complex density functions. To the best of our knowledge, it is the first work that uses OT to sample from 3D density functions.

Suppose the source probability distribution $d\mu(x) = f(x)dx$ is defined on $\Omega = [0,1]^3$ with $\mu(\Omega) = 1$. The target distribution $d\nu(y) = dy$ is the uniform distribution. We use the FFT-OT algorithm Alg. 1 to compute the OT map $T : \Omega \to \Omega, T_{\#}\mu = \nu$. The domain is tessellated to a $256 \times 256 \times 256$ grid. For each $x_{ijk}$, $i, j, k \in \{0, 1, \ldots, 255\}$, the image $T(x_{ijk})$ can be obtained. We use $\{T(x_{ijk})\}$ as vertices to compute the Delaunay triangulation of $\Omega$. Then representing the OT map $T : (\Omega, \mu) \to (\Omega, \nu)$ as a piecewise linear map, the restriction of $T$ on each tetrahedron is a linear map. Then the inverse OT map $T^{-1} : (\Omega, \nu) \to (\Omega, \mu)$ is also a piecewise linear map. Namely, given a grid point $y_{mnl}$, we can find a tetrahedron containing it. Suppose the vertices of the tetrahedron are $\{T(x_i), T(x_j), T(x_k), T(x_l)\}$, then $y_{mnl}$ is computed as

$$y_{mnl} = \lambda_i T(x_i) + \lambda_j T(x_j) + \lambda_k T(x_k) + \lambda_l T(x_l),$$

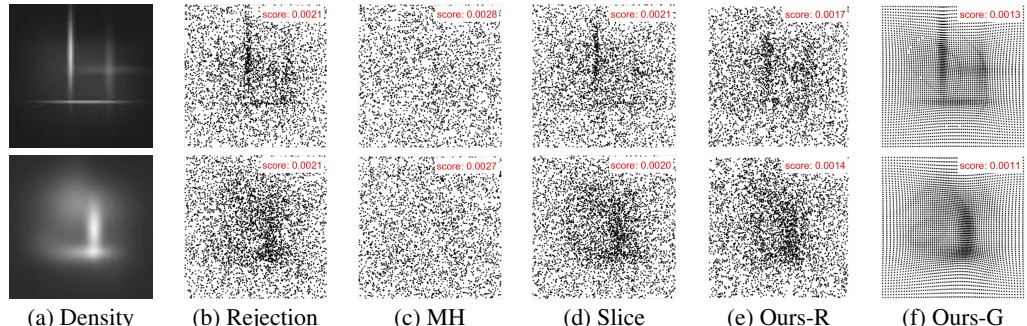

|  (a) Density | (b) Rejection | (c) MH | (d) Slice | (e) Ours-R | (f) Ours-G |

**Figure 2:** 3D density function sampling. (a) The density functions in a slice. The slices in each row come from two different density functions. (b)-(f) The samples obtained by different sampling methods. (b) Rejection sampling. (c) Metropolis-Hastings (MH) algorithm Bishop (2006). (d) Slice sampling Neal (2003). (e) The sampling results by mapping the random samples from the uniform distribution back to the desired distribution with $T^{-1}$. (f) The sampling results by mapping the grid centers back with $T^{-1}$. The scores of the top right give the results of the Chi-square goodness-of-fit test. Smaller means better.

where the non-negative barycenter coordinates satisfy $\lambda_i + \lambda_j + \lambda_k + \lambda_l = 1$. Then the image of the inverse OT map is given by

$$T^{-1}(y_{mnl}) = \lambda_i x_i + \lambda_j x_j + \lambda_k x_k + \lambda_l x_l. \tag{20}$$

We generate random samples $\{y_k\}$ according to the uniform distribution $\nu$ on $\Omega$, then their images $\{T^{-1}(y_k)\}$ are the desired random samples following the distribution $\mu$.

In our experiment, we use the same Gaussian mixture settings of the density function as Sec. 4.1. Fig. 2 visualizes the generated samples. We randomly pick the $k$-th slice along the $z$-direction from the discretized volume, draw the source density function on this slice, and use pixel intensity to represent the density in Fig. 2(a). (i) We uniformly generate $100k$ random samples $\{y_k\} \subset \Omega$, and obtain the desired random samples by applying the inverse OT map $\{T^{-1}(y_k)\}$. (ii) We also set $\{y_k\}$ as the grid centers of $\Omega$ and obtain the corresponding samples of the desired distribution $\mu$. The samples around the $k$-th slice of both sampling strategies are plotted in Fig. 2(e) and Fig. 2(f).

By visual comparison, it is obvious that the distributions of Fig. 2(e) and Fig. 2(f) are consistent with the density function in Fig. 2(a). The consistency of the boundary of Fig. 2(e) and (f) and Fig. 2(a) also verifies the obliqueness boundary condition of the Monge-Ampère equation. To further show the performance of the proposed method, we compare it with the classical sampling methods, namely rejection sampling, the Metropolis-Hastings algorithm Bishop (2006) and the slice sampling Neal (2003), shown in Fig. 2(b), Fig. 2(c) and Fig. 2(d). To quantitatively compare the sampling results, we use the Chi-square goodness-of-fit test, which firstly groups the data and then computes the $L^2$ norm of the difference between the actual number of observations in each group and the expected number of observations. In our experiment, we set the group number to $64 \times 64 \times 64$ and use 500K samples to make the comparison. The corresponding $L^2$ norm of each method is shown in the top-right of the corresponding figure. We can see that the both sampling strategies of our method give smaller scores than the classical ones.

### 4.3 VOLUMETRIC MAGNIFIER

In reality, physical magnifiers can only magnify planar images. In medical image processing, it is highly desirable to magnify certain regions of the 3D MRIs or CT images. Our algorithm can address such requests with the user prescribed region of interest (ROI) and magnifying factor. Suppose the ROI is a symmetric region with the center $(\bar{x}, \bar{y}, \bar{z}) \in \Omega$ and the radius $\sigma_x, \sigma_y, \sigma_z$ in different directions. The density function $f$ of the source measure $\mu$ is defined as

$$f(x, y, z) = 0.5 + 0.5 e^{-((x-\bar{x})^2/2\sigma_x^2 + (y-\bar{y})^2/2\sigma_y^2 + (z-\bar{z})^2/2\sigma_z^2)}$$

We compute OT map $T : (\Omega, \mu) \to (\Omega, \nu)$, where $\nu$ is the uniform distribution. Similar to the method in 3D adaptive sampling, we compute the Delaunay triangulation of the images $\{T(x_{ijk})\}$, then the OT map $T$ is represented as a piecewise linear map. The inverse optimal transportation map

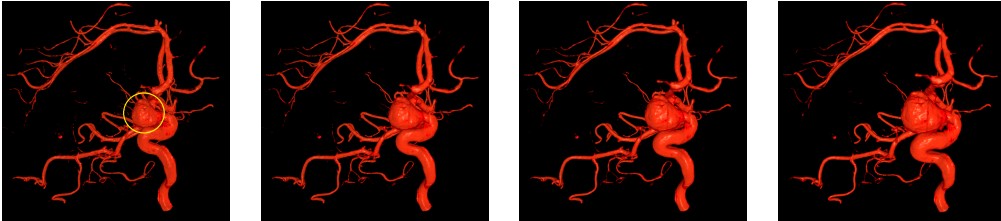

**Figure 3:** The volume magnifier of an aneurysm. The first column shows the original volumetric data, and the last three columns give the magnified data from the same viewpoints with different magnifying ratios. The yellow circle denotes the ROI/aneurysm. To obtain the results, we set $\sigma = \sigma_x = \sigma_y = \sigma_z$, and they are 0.83, 0.75 and 0.5 respectively.

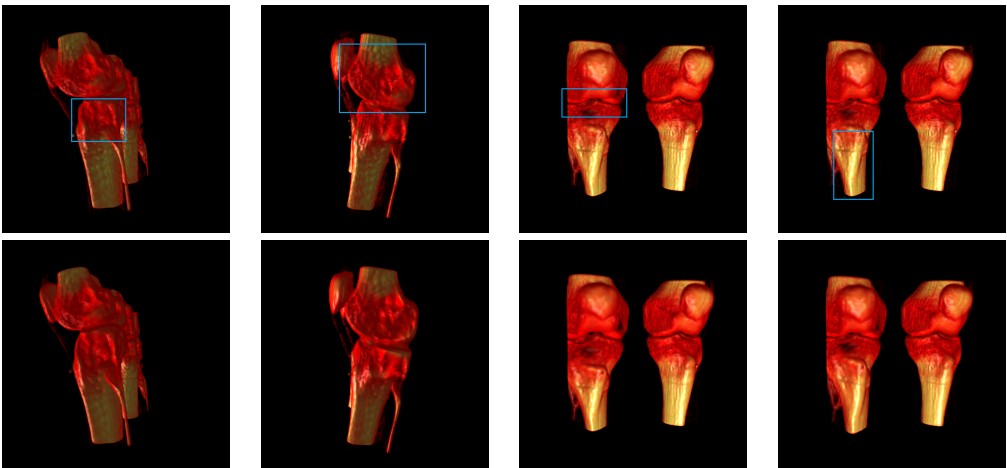

**Figure 4:** The volume magnifier of the knee. The first row gives the the original volumetric data with different ROIs denoted by the blue boxes from different viewpoints, and the second row shows the corresponding magnified results. In the experiments we set $\sigma_x = \sigma_y = \sigma_z = 0.75$.

$T^{-1} : (\Omega, \nu) \to (\Omega, \mu)$ is also piecewise linear. For each grid point $y_{mnl} \in \Omega$, we use Eqn. (20) to find its pre-image. Similarly, its corresponding intensity $I_{mnl}$ is computed by linear interpolation. Then we obtain the new volumetric data $\{I_{mnl}\}$ with the magnified ROI and visualize the result with Voreen Meyer-Spradow et al. (2009).

Fig. 3 demonstrates our volumetric magnifier by magnifying an aneurysm on blood vessel Hansen & Johnson (2004). We choose the aneurysm region as the ROI. The first column gives the snapshot of the blood vessel, and the yellow circle denotes the location of the aneurysm. The last three columns show the magnified aneurysm with different magnifying ratio from the same viewpoints. Moreover, we show the magnified volumetric knee from different viewpoints with different ROIs denoted by the blue boxes in Fig. 4. Our method only magnifies the ROIs and keeps other regions unchanged. Compared with the traditional method requiring tedious zoom in/out, our method only magnifies the ROI region and keeps the whole subject in the field of view, which enables doctors to visualize the overall anatomy while scrutinize detailed anatomical structure at the same time.

## 5 Conclusion

In this paper, we propose the FFT-OT method to solve the optimal transportation problem. According to the Brenier theory, under the quadratic distance cost, finding the solution to the OT problem is equivalent to solving the Monge-Ampère equation, which can be linearized as a sequence of variant coefficient elliptic PDEs. Later, the variant coefficient PDEs are approximated by constant coefficient PDEs and solved by Fast Fourier Transformation. We also prove that the proposed method converges linearly. Experiments on volumetric data show that the FFT-OT can be used to sample from complex 3D density functions and magnify the volumetric data in medical images.

ACKNOWLEDGEMENT

This research was partially supported by National Key R&D Program of China 2021YFA1003003 and NSFC No. 61936002, T2225012. This work was also partially supported by NIH 3R01LM012434-05S1, 1R21EB029733-01A1, NSF FAIN-2115095 and NSF CMMI-1762287.

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

## A    RELATED WORK

There is a huge literature about optimal transportation. Here we will only briefly review the most related works. For detailed reviews, we refer readers to Santambrogio (2015); Peyré & Cuturi (2019).

The first type of algorithms is based on the Kantorovich theory. When both the input and output domains are Dirac masses, the Kantorovich problem can be treated as a standard linear programming (LP) task. In order to tackle large data sets, Cuturi (2013) adds an entropic regularizer to the original LP problem and the regularized problem can be quickly solved by the Sinkhorn algorithm. Recently, various algorithms have been proposed to further accelerate the computation by improving the efficiency of matrix-vector multiplications, including the Greenkhorn Altschuler et al. (2017), Screenkhorn Alaya et al. (2019) and the NYS-SINK Altschuler et al. (2019) algorithms. Dvurechensky et al. Dvurechensky et al. (2018) also propose the adaptive primal-dual accelerated gradient descent algorithm (APDAGD) to solve the discrete OT problem. An et al. An et al. (2022) compute the approximate OT plan by smoothing the dual Kantorovich problem and solving it with the FISTA method. This kind of methods have limitations: (i) they only give transport plans and cannot produce the bijective transportation maps; and (ii) the computational complexity is too high to apply them in the scenarios with huge number of samples.

The second type of algorithms is based on the Brenier theory Brenier (1987) and its intrinsic connection with convex geometry Gu et al. (2016). The semi-discrete OT algorithm proposed in Aurenhammer et al. (1998) finds the transport map between a continuous distribution and a discrete measure via a variational approach by dynamically constructing the power diagrams. Its efficiency can be further improved Levy (2015); Merigot (2011) by the multi-resolution strategy. The algorithms proposed in Kitagawa et al. (2019); Su et al. (2017) also improve the efficiency by applying the Newton's method. When both the source and target measures are continuous, some interpolation methods are necessary Schwartzburg et al. (2014). The major drawback of this type of algorithms is the high computational complexity of constructing the dynamic power diagram, which prevents them from handling high dimensional tasks. For example, for the 3D OT problems, these algorithms usually run very slow.

The third type of algorithms is based on computational fluid dynamics Benamou et al. (2002); Papadakis et al. (2014). These methods aim at finding a special temporal-spacial flow field that transports the initial source density to the target density with the minimal total kinetic energy. Then the diffeomorphism induced by the flow gives the optimal transport map under the quadratic Euclidean distance cost. However, this kind of algorithms are difficult to extend to high dimensional space.

The fourth type of algorithms directly solve the Monge-Ampère equation using numerical methods. Loeper and Rapetti Loeper & Rapetti (2005) propose to solve the linearized Monge-Ampère equation defined on a flat torus in each iteration. Its corresponding variant coefficient elliptic PDE is converted to a positive definite linear system using the finite-difference scheme, which can be solved by the BiCG algorithm Endre (2020). Benamou et al. Benamou et al. (2014) propose to solve the linearized Monge-Ampère on more general domains using Newton's method. Nader and Guennebaud Nader & Guennebaud (2018) apply the similar discretization strategy and solve the Monge-Ampère equation by conjugate gradient method. Saumier et al. Saumier et al. (2013) propose to solve the linearized Monge-Ampère equation using FFT. In each iteration the elliptic PDE with spacial and temporal variant coefficients is converted to a group of linear equations in the frequency domain, which is solved by the GMRES algorithm. Although the GMRES algorithm can be implemented on GPUs Aliaga et al. (2019), there is no available open source code. The work in Saumier et al. (2013) focuses on periodic boundary condition, but this our proposed work focuses on general second boundary condition; the work in Saumier et al. (2013) concerns planar OT maps, ours emphasizes on volumetric OT maps, which has higher complexity. The work in Saumier et al. (2013) can handle more general target measures, the proposed work currently only deals with the Lebesgue target measure. Nevertheless, the current work can be directly generalized to handle general target measures as well. Lei and Gu Lei & Gu (2021) use the fixed point method to compute the 2-dimensional OT problem based on FFT, but it cannot be extended to solve the 3-dimensional problems.

In this work, we combine the idea of linearizing the Monge-Ampère equation Loeper & Rapetti (2005) and the idea of FFT Saumier et al. (2013). The key novelty of our proposed method is to use the *mean* linearized Monge-Ampère operator Eqn. (12) to replace the conventional linearized

Monge-Ampere operator Eqn. (10). This replacement allows the algorithm to be implemented on GPUs and makes the algorithm hundreds of times faster. In the following, we compute the 3-dimensional optimal transport problem by applying the proposed algorithm. Our method also runs more than $100\times$ faster than the convex geometry based method Levy (2015).

# B APPENDIX THEORY

In the section, we give the detailed proofs for several lemmas and theorems. Some of them are well known in the Monge-Ampère PDE field and the applied mathematics field, we include them for the completeness.

## B.1 EXISTENCE OF THE SOLUTION TO THE TIME DEPENDENT MONGE-AMPÈRE EQNUATION

Let $\mathbb{T}^n = \mathbb{R}^n/\mathbb{Z}^n$ be the $n$-dimensional flat torus. Below we sometimes identify it with $\Omega = [0,1]^n$ and assume all data are periodic. The existence and regularity of solutions to the Monge-Ampère equation are given by the following theorem,

**Theorem 5.** *Suppose a positive density function $f : \Omega \to \mathbb{R}$ is defined on $\Omega = [0,1]^n$, such that $\int_\Omega f(x)dx = 1$, and $f \in C^\alpha(\Omega)$, then the solution $u : \Omega \times [0,1]$ to the time-dependent Monge-Ampère equation*

$$det D_x^2 u(x,t) = (1-t) + tf(x), \quad \nabla_x u(x,t)(\Omega) = \Omega \tag{21}$$

*exists and is unique up to a constant. Furthermore, there exist constants $0 < \lambda < \Lambda$, such that*

$$\lambda \sum_{p=1}^n \xi_p^2 \leq \sum_{p,q=1}^n u^{pq}(x,t)\xi_p\xi_q \leq \Lambda \sum_{p=1}^n \xi_p^2, \quad \forall \xi \in \mathbb{R}^n, \ \forall (x,t) \in \Omega \times [0,1]. \tag{22}$$

We refer readers to Cordero-Erausquin (1999) for detailed proof.

**Weak Solution** In practice, we compute the weak solution of the linearized Monge-Ampère Eqn. (6) using numerical methods. We first rewrite the differential operator to a divergence form, then define a bi-linear form.

Since $(u^{pq}(x,t))$ is the adjoint matrix of $D_x^2 u(x,t)$, by direct computation, we obtain

$$\sum_{p=1}^n \partial_p u^{pq}(x,t) = 0, \quad \forall (x,t) \in \Omega \times [0,1], \quad \forall q = 1,\ldots,n. \tag{23}$$

so Eqn. (6) can be converted into the divergence form:

$$\sum_{p=1}^n \partial_p \left( \sum_{q=1}^n u^{pq} \partial_q v \right) = \sum_{p,q=1}^n u^{pq} \partial_p \partial_q v + \sum_{q=1}^n \left( \sum_{p=1}^n \partial_p u^{pq} \right) \partial_q v = \sum_{p,q=1}^n u^{pq} \partial_p \partial_q v,$$

we obtain

$$\sum_{p=1}^n \partial_p \left( \sum_{q=1}^n u^{pq}(x,t) \partial_q v(x,t) \right) = f(x) - 1. \tag{24}$$

with Neumann boundary condition

$$\frac{\partial v(x,t)}{\partial \mathbf{n}} = 0, \quad \forall (x,t) \in \partial\Omega \times [0,1]. \tag{25}$$

For any $w \in H^1(\Omega)$, by differentiation of product, we obtain

$$\sum_{p=1}^n \partial_p \left( \sum_{q=1}^n u^{pq} \partial_q v \right) w + \sum_{p=1}^n \left( \sum_{q=1}^n u^{pq} \partial_q v \right) \partial_p w = \sum_{p=1}^n \partial_p \left[ \left( \sum_{q=1}^n u^{pq} \partial_q v \right) w \right]$$

by integrating both sides, and from the fact that $v$ satisfies the Neumann boundary condition, we deduce

$$\int_\Omega \sum_{p=1}^n \partial_p \left( \sum_{q=1}^n u^{pq} \partial_q v \right) w + \int_\Omega \sum_{p,q=1}^n u^{pq} \partial_q v \partial_p w = \int_{\partial\Omega} \sum_{p=1}^n \left( \sum_{q=1}^n u^{pq} \partial_q v \right) w = 0. \tag{26}$$

For any fixed time $t \in [0, 1]$, by the divergence form, we can construct a bilinear form $a : H^1(\Omega) \times H^1(\Omega)$ and a linear form $l : H^1(\Omega) \to \mathbb{R}$,

$$a(v, w) = \sum_{p,q=1}^{n} \int_{\Omega} u^{pq} \partial_p v \partial_q w, \quad l(w) = -\int_{\Omega} (f - 1) w dx. \tag{27}$$

A *weak solution* to Eqn. (24) is a function $v \in H^1(\Omega)$, such that

$$a(v, w) = l(w), \quad \forall w \in H^1(\Omega). \tag{28}$$

By the uniform ellipticity Eqn. (22), the Lax-Milgram theorem Endre (2020) shows the existence of the weak solution.

## B.2 Discrete Linearized Monge-Ampère Equation Solvability

**Galerkin Method** In practice, we construct a triangulation $\mathcal{T}$ of $\Omega$, such that the ratio between the diameter and inscribe-sphere radius of each simplex is bounded, and variation of the diameters of all the simplexes is small. We call such kind of $\mathcal{T}$ a *quasi-uniform* triangulation, and denote the largest diameter as $h$. For each vertex $v_i \in \mathcal{T}$, we construct a piecewise linear base function $\varphi_i$, such that $\varphi_i$ is linear on each triangle, $\varphi_i(v_j)$ is $\delta_{ij}$. We define a finite dimensional subspace $V_h \subset H^1(\Omega)$,

$$V_h := \left\{ v_h(x) := \sum_{v_i \in \mathcal{T}} \lambda_i \varphi_i(x), \lambda_i \in \mathbb{R} \right\}.$$

Given a function $u \in H^1(\Omega)$, we use $u_h \in V_h$ to denote its approximation in $V_h$. Furthermore, $u_h = \sum_i \lambda_i \varphi_i$, we also use $u_h$ to represent the coefficient vector $(\lambda_1, \lambda_2, \ldots, \lambda_k)^T$ depending on the context. The weak solution Eqn. (28) to the Monge-Ampère equation (6) is equivalent to find a $v \in H^1(\Omega)$, such that $a(v, w) = l(w)$ for all $w \in H^1(\Omega)$. In discrete cases, we want to find $v_h \in V_h$, such that

$$a(v_h, w_h) = l(w_h), \quad \forall w_h \in V_h. \tag{29}$$

Eqn. (29) is equivalent to the linear system,

$$\begin{pmatrix} a(\varphi_1, \varphi_1) & a(\varphi_2, \varphi_1) & \cdots & a(\varphi_N, \varphi_1) \\ a(\varphi_1, \varphi_2) & a(\varphi_2, \varphi_2) & \cdots & a(\varphi_N, \varphi_2) \\ \vdots & \vdots & & \vdots \\ a(\varphi_1, \varphi_N) & a(\varphi_2, \varphi_N) & \cdots & a(\varphi_N, \varphi_N) \end{pmatrix} \begin{pmatrix} \lambda_1 \\ \lambda_2 \\ \vdots \\ \lambda_N \end{pmatrix} = \begin{pmatrix} l(\varphi_1) \\ l(\varphi_2) \\ \vdots \\ l(\varphi_N) \end{pmatrix} \tag{30}$$

From the weak solution to the linearized Monge-Ampère equation (10), we obtain the linear system Eqn. (30). We denote the *stiffness matrix* $A = (a(\varphi_i, \varphi_j))$. By the uniform ellipticity Eqn. (22), and $V_h \subset H^1(\Omega)$

$$a(v, v) \geq \lambda \|\nabla v\|_{L^2(\Omega)}^2$$

Assume $\int_{\Omega} v dx = 0$, by Poincaré inequality,

$$\|\nabla v\|_{L^2(\Omega)}^2 \geq C_1(\Omega) \|v\|_{L}^2(\Omega), \quad \forall v \in H^1(\Omega), \int_{\Omega} v dx = 0,$$

where the constant $C_1(\Omega)$ depends on $\Omega$. Combine the above two inequalities, we obtain

$$a(v, v) \geq c \|v\|_{L^2(\Omega)}^2, \quad \forall v \in H^1(\Omega), \int_{\Omega} v dx = 0. \tag{31}$$

Similarly, By the uniform ellipticity Eqn. 22, and $V_h \subset H^1(\Omega)$

$$a(v, v) \leq \Lambda \|\nabla v\|_{L^2(\Omega)}^2$$

For linear finite element and quasi-uniform triangulation, we have the inverse Poincaré inequality,

$$\|\nabla v_h\|_{L^2}^2 \leq C_2(\Omega) h^{-1} \|v_h\|_{L^2}^2.$$

where $h$ is the diameter of each element. Combine the above two inequalities, we obtain

$$a(v_h, v_h) \leq C \|v_h\|^2_{L^2(\Omega)}, \quad \forall v_h \in V_h. \tag{32}$$

By combining the inequalities Eqn. (31) and Eqn. (32), we obtain

$$\frac{1}{C_3} \|v_h\|^2_{L^2(\Omega)} \leq a(v_h, v_h) \leq C_3 \|v_h\|^2_{L^2(\Omega)}, \quad \forall v_h \in V_h, \int_\Omega v_h = 0, \tag{33}$$

where $C_3 > 1$ is a constant. Suppose $v_h = \sum_{i=1}^n \xi_i \varphi_i$, then

$$\|v_h\|^2_{L^2(\Omega)} = \int_\Omega v_h^2 dx = \sum_{i,j=1}^n \xi_i \xi_j \int_\Omega \varphi_i(x) \varphi_j(x) dx = \xi^T \Phi \xi,$$

where $\xi = (\xi_i)$ and the matrix $\Phi = \left( \int_\Omega \varphi_i \varphi_j \right)$ is positive definite. Therefore,

$$\frac{1}{C_4} \|\xi\|^2 \leq \xi^T \Phi \xi < C_4 \|\xi\|^2. \tag{34}$$

By $a(v_h, v_h) = \xi^T A \xi$, combing inequalities Eqn. (33) and Eqn. (34), we obtain

$$\frac{1}{C_3 C_4} \|\xi\|^2 \leq \xi^T A \xi \leq C_3 C_4 \|\xi\|^2, \quad \forall \xi \in \mathbb{R}^n, \sum_{i=1}^n \xi_i = 0, \tag{35}$$

where $C_3 C_4 > 1$. This proves the following lemma,

**Lemma 6.** *By using Galerkin method using linear elements to numerically approximate the weak solution Eqn. (28) to the linearized Monge-Ampère Eqn. (6), if the uniform ellipticity Eqn. (22) holds, and the triangulation $\mathcal{T}$ is quasi-uniform, then the stiffness matrix of the linear system Eqn. (30) is positive definite on the space $\sum_{i=1}^n \xi_i = 0$,*

$$\frac{1}{C_3 C_4} \|\xi\|^2 \leq \xi^T A \xi \leq C_3 C_4 \|\xi\|^2, \quad \forall \xi \in \mathbb{R}^n, \sum_{i=1}^n \xi_i = 0, \tag{36}$$

*where $C_3 C_4 > 1$.*

Since the uniform ellipticity Eqn. (22) holds for any time $t \in [0, 1]$, then we obtain

**Corollary 7.** *By using Galerkin method with linear elements on quasi-uniform triangulations, the linearized Monge-Ampère equation in the continuity method Eqn. (6) always has a solution $v_h \in V_h$ for any $t \in [0, 1]$.*

Please note that the central differential scheme can be treated as Galerkin's method on a special uniform triangulation. Therefore, the above estimates still hold.

## B.3 CONVERGENCE RATE

**Theorem 8** (main). *Given a domain $\Omega \subset \mathbb{R}^n$, which is a canonical cuboid $\Omega = [-1, 1]^n$, and a positive density function $f : \Omega \to \mathbb{R}$ with the balance condition*

$$\int_\Omega f(x) dx = \int_\Omega 1 \cdot dx,$$

*suppose the mirror reflection extension Eqn. (14) of $f$ to the flat torus $\tilde{f} : \mathbb{T}^n \to \mathbb{R}$ is $C^\alpha$, $\alpha \in (0, 1)$, then Monge-Ampère equation,*

$$det D^2 u(x) = f(x), \quad \nabla u(\Omega) = \Omega$$

*can be solved using FFT-OT Algorithm Alg. (1). In particular, one can choose the step length parameter $\tau$, such that there is a constant $0 < \gamma < 1$, the approximation error satisfies*

$$\|f - \rho_{k+1}\|^2 < C\gamma^k,$$

*namely the algorithm has a linear convergence rate.*

*Proof.* Suppose at the $k + 1$-th iteration, $\rho_{k+1} = \det(I + D^2 u_{k+1})$, $\|v_k\| \sim O(\tau^{-1})$,

$$
\begin{aligned}
f - \rho_{k+1} &= f - \det(I + \mathcal{D}^2 u_k + \mathcal{D}^2 v_k) \\
&= f - \det(I + \mathcal{D}^2 u_k) - \sum_{pq} u_k^{pq} \partial_p \partial_q v_k + o(\tau^{-1}) \\
&= (f - \rho_k) - L_k[v_k] + o(\tau^{-1})
\end{aligned}
$$

where $L_k[v_k] = \sum_{pq} u_k^{pq} \partial_p \partial_q v_k$. Hence by integration by parts Eqn. (27),

$$
\begin{aligned}
\|f - \rho_{k+1}\|_{L^2(\Omega)}^2 &= \|f - \rho_k\|_{L^2(\Omega)}^2 - 2 \int_\Omega L_k[v_k](f - \rho_k) + o(\tau^{-1}) \\
&= \|f - \rho_k\|_{L^2(\Omega)}^2 + 2a_k(f - \rho_k, v_k) + o(\tau^{-1})
\end{aligned}
$$

where $a_k$ is the bilinear form in Eqn.(27). In the discrete case, all functions are in $V_h$, we denote

$$
\|u_h\|_\Phi^2 := \|u_h\|_{L^2(\Omega)}^2 = u_h^T \Phi u_h, \quad \|u_h\|^2 := u_h^T u_h, \quad \|u_h\|_A^2 := u_h^T A u_h,
$$

by the inequality Eqn. (34) and Eqn. 35,

$$
\frac{1}{C_4} \|u_h\|^2 \le \|u_h\|_\Phi^2 \le C_4 \|u_h\|^2, \quad \frac{1}{C_3 C_4} \|u_h\|^2 \le \|u_h\|_A^2 \le C_3 C_4 \|u_h\|^2.
$$

Therefore

$$
\|f_h - \rho_{h,k+1}\|_\Phi^2 = \|f_h - \rho_{h,k}\|_\Phi^2 - 2\tau^{-1}(f - \rho_{h,k})^T A_k \bar{A}_k^{-1}(f_h - \rho_{h,k}) + o(\tau^{-1}), \tag{37}
$$

where $A_k$ is the stiffness matrix in Eqn.(30), and $\bar{A}_k$ is the mean stiffness matrix. ( By the uniform ellipticity Eqn. (22), the eigen values of the adjoint matrix $(u^{pq})(x, t)$ is uniformly bounded away from zero in the space $\mathcal{H} := \{\xi \in \mathbb{R}^n | \sum_i \xi_i = 0\}$, so the eigen value of the mean adjoint matrix $\bar{u}^{pq}(t)$ is bounded away from zero in $\mathcal{H}$. After discretization, the eigen values of $\bar{A}_k$ is strictly positive in $\mathcal{H}$, hence $\bar{A}_k$ is invertible in $\mathcal{H}$. In the following discussion, the term $o(\tau^{-1})$ will be ignored.) Remark that the following displayed equation is a scalar

$$
(f_h - \rho_{h,k})^T A_k \bar{A}_k^{-1}(f - \rho_{h,k}) = \mathrm{tr}((f_h - \rho_{h,k})^T A_k \bar{A}_k^{-1}(f_h - \rho_{h,k}))
$$

Since $A_k$ and $\bar{A}_k$ are symmetric, positive definite on the space $\sum_i \xi_i = 0$, $\|A_k\|_2 \le C_3 C_4$ and $\|\bar{A}_k\|_2 \le C_3 C_4$, so are their inverses. Since $A_n$ and $\bar{A}_n$ are symmetric, positive definite on the space orthogonal to $(1, 1, \ldots, 1)^T$, by Eqn. (35) and $\|A_k \bar{A}_k^{-1}\| \le \|A_k\|\|\bar{A}_k^{-1}\|$, we have

$$
\frac{(n-1)}{C_3^2 C_4^3} \|f_h - \rho_{h,k}\|_\Phi^2 \le (f_h - \rho_{h,k})^T A_k \bar{A}_k^{-1}(f_h - \rho_{h,k}).
$$

Plug into Eqn. (37), we have

$$
\|f_h - \rho_{h,k+1}\|_\Phi^2 \le \left(1 - \frac{1}{\tau}\frac{(n-1)}{C_3^2 C_4^3}\right) \|f_h - \rho_{h,k}\|_\Phi^2 \le \left(1 - \frac{1}{\tau}\frac{(n-1)}{C_3^2 C_4^3}\right)^k \|f_h - \rho_{h,0}\|_\Phi^2. \tag{38}
$$

We can choose the step-length $\tau^{-1}$, such that $\gamma \in (0, 1)$, where

$$
\gamma = 1 - \frac{(n-1)}{\tau C_3^2 C_4^3}.
$$

Therefore

$$
\|f_h - \rho_{h,k+1}\|_\Phi^2 \le \gamma^k \|f_h - \rho_{h,0}\|_\Phi^2 \le C_4 \gamma^k \|f_h - \rho_{h,0}\|^2. \tag{39}
$$

$\square$

## B.4 Differential Operator Using FFT

By using the Discrete Fourier Transformation, the differential operators can be converted to algebraic operators in the frequency domain.

**Lemma 9.** *Suppose the discrete function is $u_{i,j,k}$, with discrete Fourier transformation*

$$u_{i,j,k} = \frac{1}{MNL} \sum_{m=0}^{M-1} \sum_{n=0}^{N-1} \sum_{l=0}^{L-1} \hat{u}_{m,n,l} e^{\sqrt{-1}\frac{2\pi mi}{M}} e^{\sqrt{-1}\frac{2\pi nj}{N}} e^{\sqrt{-1}\frac{2\pi lk}{L}}$$

*then the differential operator using central difference $\partial_i \partial_i u_{i,j,k}$ is given by*

$$\partial_i \partial_i u_{i,j,k} = \frac{1}{h_x^2}(u_{i+1,j,k} + u_{i-1,j,k} - 2u_{i,j,k})$$

$$= \frac{1}{MNL} \sum_{m=0}^{M-1} \sum_{n=0}^{N-1} \sum_{l=0}^{L-1} \hat{u}_{m,n,l} \frac{2\left(\cos\frac{2\pi m}{M} - 1\right)}{h_x^2} e^{\iota\frac{2\pi mi}{M}} e^{\iota\frac{2\pi nj}{N}} e^{\iota\frac{2\pi lk}{L}}$$

*where $\iota = \sqrt{-1}$, and $\partial_i \partial_j u_{i,j,k}$ is given by,*

$$\partial_i \partial_j u_{i,j,k} = \frac{1}{4h_x h_y}(u_{i+1,j+1,k} + u_{i-1,j-1,k} - u_{i+1,j-1,k} - u_{i-1,j+1,k})$$

$$= \frac{1}{MNL} \sum_{m=0}^{M-1} \sum_{n=0}^{N-1} \sum_{l=0}^{L-1} \hat{u}_{m,n,l} \frac{-\sin\frac{2\pi m}{M}\sin\frac{2\pi n}{N}}{h_x h_y} e^{\iota\frac{2\pi mi}{M}} e^{\iota\frac{2\pi nj}{N}} e^{\iota\frac{2\pi lk}{L}}$$

*Proof.* By equations

$$\cos(A + \alpha) + \cos(A - \alpha) - 2\cos(A)$$
$$= (\cos A \cos\alpha - \sin A \sin\alpha) + (\cos A \cos\alpha + \sin A \sin\alpha) - 2\cos A$$
$$= 2(\cos\alpha - 1)\cos A$$

and

$$\sin(A + \alpha) + \sin(A - \alpha) - 2\sin(A)$$
$$= (\sin A \cos\alpha + \cos A \sin\alpha) + (\sin A \cos\alpha - \cos A \sin\alpha) - 2\cos A$$
$$= 2(\cos\alpha - 1)\sin A$$

we obtain

$$\frac{1}{h_x^2}[e^{\iota\frac{2\pi m(i+1)}{M}} + e^{\iota\frac{2\pi m(i-1)}{M}} - 2e^{\iota\frac{2\pi mi}{M}}] = \frac{2\left(\cos\frac{2\pi m}{M} - 1\right)}{h_x^2}e^{\iota\frac{2\pi mi}{M}}$$

by direct computation, we have

$$\partial_i \partial_i u_{i,j,k} = \frac{1}{h_x^2}(u_{i+1,j,k} + u_{i-1,j,k} - 2u_{i,j,k})$$

$$= \frac{1}{MNL} \sum_{m=0}^{M-1} \sum_{n=0}^{N-1} \sum_{l=0}^{L-1} \hat{u}_{m,n,l} \frac{e^{\iota\frac{2\pi m(i+1)}{M}} + e^{\iota\frac{2\pi m(i-1)}{M}} - 2e^{\iota\frac{2\pi mi}{M}}}{h_x^2} e^{\iota\frac{2\pi nj}{N}} e^{\iota\frac{2\pi lk}{L}}$$

$$= \frac{1}{MNL} \sum_{m=0}^{M-1} \sum_{n=0}^{N-1} \sum_{l=0}^{L-1} \hat{u}_{m,n,l} \frac{2\left(\cos\frac{2\pi m}{M} - 1\right)}{h_x^2} e^{\iota\frac{2\pi mi}{M}} e^{\iota\frac{2\pi nj}{N}} e^{\iota\frac{2\pi lk}{L}}$$

Similarly, by equations

$$\cos(A + \alpha + B + \beta) + \cos(A - \alpha + B - \beta) - \cos(A + \alpha + B - \beta) - \cos(A - \alpha + B + \beta)$$
$$= \cos(A + B + \alpha + \beta) + \cos(A + B - \alpha - \beta) - \cos(A + B + \alpha - \beta) - \cos(A + B - \alpha + \beta)$$
$$= 2\cos(A + B)\cos(\alpha + \beta) - 2\cos(A + B)\cos(\alpha - \beta)$$
$$= 2\cos(A + B)(\cos(\alpha + \beta) - \cos(\alpha - \beta))$$
$$= 2\cos(A + B)(\cos\alpha\cos\beta - \sin\alpha\sin\beta - \cos\alpha\cos\beta - \sin\alpha - \sin\beta)$$
$$= -4\cos(A + B)\sin\alpha\sin\beta$$

and

$$\sin(A + \alpha + B + \beta) + \sin(A - \alpha + B - \beta) - \sin(A + \alpha + B - \beta) - \sin(A - \alpha + B + \beta)$$
$$= \sin(A + B + \alpha + \beta) + \sin(A + B - \alpha - \beta) - \sin(A + B + \alpha - \beta) - \sin(A + B - \alpha + \beta)$$
$$= 2\sin(A + B)\cos(\alpha + \beta) - 2\sin(A + B)\cos(\alpha - \beta)$$
$$= 2\sin(A + B)(\cos(\alpha + \beta) - \cos(\alpha - \beta))$$
$$= 2\sin(A + B)(\cos\alpha\cos\beta - \sin\alpha\sin\beta - \cos\alpha\cos\beta - \sin\alpha - \sin\beta)$$
$$= -4\sin(A + B)\sin\alpha\sin\beta$$

we deduce the following equation,

$$\partial_i \partial_j u_{i,j,k} = \frac{1}{4h_x h_y}(u_{i+1,j+1,k} + u_{i-1,j-1,k} - u_{i+1,j-1,k} - u_{i-1,j+1,k})$$

$$= \frac{1}{MNL} \sum_{m=0}^{M-1} \sum_{n=0}^{N-1} \sum_{l=0}^{L-1} \hat{u}_{m,n,l} \frac{-\sin\frac{2\pi m}{M}\sin\frac{2\pi n}{N}}{h_x h_y} e^{\iota\frac{2\pi mi}{M}} e^{\iota\frac{2\pi nj}{N}} e^{\iota\frac{2\pi lk}{L}}$$

$\square$

Similarly, we have the representations of other differential operators in the frequency domain,

$$\partial_j \partial_j u_{i,j,k} = \frac{1}{h_x^2}(u_{i,j+1,k} + u_{i,j-1,k} - 2u_{i,j,k})$$

$$= \frac{1}{MNL} \sum_{m=0}^{M-1} \sum_{n=0}^{N-1} \sum_{l=0}^{L-1} \hat{u}_{m,n,l} \frac{2\left(\cos\frac{2\pi n}{N} - 1\right)}{h_y^2} e^{\iota\frac{2\pi mi}{M}} e^{\iota\frac{2\pi nj}{N}} e^{\iota\frac{2\pi lk}{L}}$$

$$\partial_k \partial_k u_{i,j,k} = \frac{1}{h_z^2}(u_{i,j,k+1} + u_{i,j,k-1} - 2u_{i,j,k})$$

$$= \frac{1}{MNL} \sum_{m=0}^{M-1} \sum_{n=0}^{N-1} \sum_{l=0}^{L-1} \hat{u}_{m,n,l} \frac{2\left(\cos\frac{2\pi l}{L} - 1\right)}{h_z^2} e^{\iota\frac{2\pi mi}{M}} e^{\iota\frac{2\pi nj}{N}} e^{\iota\frac{2\pi lk}{L}}$$

$$\partial_j \partial_k u_{i,j,k} = \frac{1}{4h_y h_z}(u_{i,j+1,k+1} + u_{i,j-1,k-1} - u_{i,j+1,k-1} - u_{i,j-1,k+1})$$

$$= \frac{1}{MNL} \sum_{m=0}^{M-1} \sum_{n=0}^{N-1} \sum_{l=0}^{L-1} \hat{u}_{m,n,l} \frac{-\sin\frac{2\pi n}{N}\sin\frac{2\pi l}{L}}{h_y h_z} e^{\iota\frac{2\pi mi}{M}} e^{\iota\frac{2\pi nj}{N}} e^{\iota\frac{2\pi lk}{L}}$$

$$\partial_k \partial_i u_{i,j,k} = \frac{1}{4h_z h_x}(u_{i+1,j,k+1} + u_{i-1,j,k-1} - u_{i+1,j,k-1} - u_{i-1,j,k+1})$$

$$= \frac{1}{MNL} \sum_{m=0}^{M-1} \sum_{n=0}^{N-1} \sum_{l=0}^{L-1} \hat{u}_{m,n,l} \frac{-\sin\frac{2\pi l}{L}\sin\frac{2\pi m}{M}}{h_z h_x} e^{\iota\frac{2\pi mi}{M}} e^{\iota\frac{2\pi nj}{N}} e^{\iota\frac{2\pi lk}{L}}$$

## C   ALGORITHM PIPELINES

In this section, we give the algorithm pipeline of the FFT-OT in Alg. 1 and the details to solve the costant coefficient elliptic PDE through FFT in Alg. 2.

---
**Algorithm 1:** FFT-OT
---

**Input:** Domain $\Omega = [-1, 1]^3$, the source density function $f > 0$, the target density $g = 1$, step length $\tau$, approximation error threshold $\varepsilon$

**Output:** Solution $\frac{1}{2}\|x\|^2 + u_n$ to the Monge-Ampère Eqn. (2) with the corresponding boundary condition.

    Initialize $u_0(x) = 0$;

    **while** *true* **do**

        Compute the Hessian matrix $D^2 u_n(x)$;

        Compute the density function $\rho_n(x) \leftarrow \det(I + D^2 u_n(x))$;

        **if** $\|f - \rho_n\|_{L_2(\Omega)} < \varepsilon$ **then**

            Break;

        Compute the adjoint matrix $[H_n^{pq}(x)] \leftarrow \text{Adj}(I + D^2 u_n(x))$;

        Compute the mean adjoint matrix $[\bar{H}_n^{pq}]$ using Eqn. (11);

        Solve the constant coefficient elliptic PDE (12) using the FFT Solver Alg. 2;

        Update the Brenier potential $u_{n+1}(x) \leftarrow u_n + \tau v_n$;

---

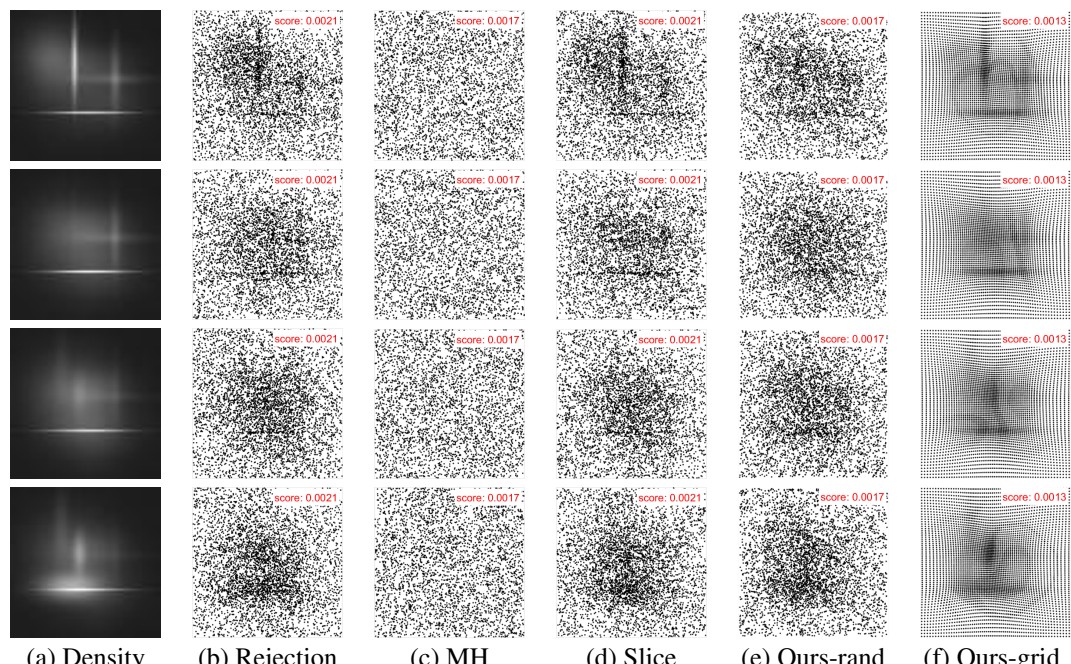

|  (a) Density | (b) Rejection | (c) MH | (d) Slice | (e) Ours-rand | (f) Ours-grid |

**Figure 5:** 3D density function sampling. (a) The density functions in different slices of the same model, namely the 40th, 56th, 72th and 80th. (b)-(f) The samples obtained by different sampling methods. (b) Rejection sampling. (c) Metropolis-Hastings (MH) algorithm Bishop (2006). (d) Slice sampling Neal (2003). (e) The sampling results by mapping the random samples from the uniform distribution back to the desired distribution with $T^{-1}$. (f) The sampling results by mapping the grid centers back with $T^{-1}$. The scores of the top right give the results of the Chi-square goodness-of-fit test. Smaller means better. Zoom in for better visualization.

---

**Algorithm 2:** FFT Solver for the Constant Coefficient Elliptic PDE

---

**Input:** Domain $\Omega = [-1, 1]^3$, $M, N, L$, $\{a^{pq}\}$, $b^r$, $c$, function $f$ with the periodic boundary condition
**Output:** Solution $u$ to the elliptic PDE Eqn. (18)

    Discretize the domain $\Omega$ to a $M \times N \times L$ grid;
    Sample the function $f$ to $f_{i,j,k}$;
    Compute FFT using Eqn. (16), $\{\hat{f}_{m,n,l}\} \leftarrow \text{FFT}(\{f_{i,j,k}\})$;
    **for** $(m, n, l) \in [0, M-1] \times [0, N-1] \times [0, L-1]$ **do**
        Compute the factor $\lambda_{m,n,l}$ using Eqn. (19);
        **if** $\lambda_{m,n,l}$ *is* 0 **then**
            $\hat{u}_{m,n,l} \leftarrow 0$;
        **else**
            $\hat{u}_{m,n,l} \leftarrow \hat{f}_{m,n,l}/\lambda_{m,n,l}$;

    Compute the Inverse FFT using Eqn. (17), $\{u_{i,j,k}\} \leftarrow \text{IFFT}(\{\hat{u}_{m,n,l}\})$;
    Return $\{u_{i,j,k}\}$.

---

## D APPENDIX EXPERIMENTS

In this section, as a compensation of the experiments in the main paper, we give more results on the 3D adaptive sampling and volumetric magnifier.

### D.1 MORE RESULTS ON 3D ADAPTIVE SAMPLING

In the experiments, we set the density function $f(x) = \sum_{i=1}^{30} p_i \mathcal{N}(\mu_i, \Sigma_i)$, where $\mathcal{N}(\mu_i, \Sigma_i)$ represents Gaussian distribution with mean $\mu_i$ and variance $\Sigma_i = \text{diag}(\sigma_{i0}^2, \sigma_{i1}^2, \sigma_{i2}^2)$. $\mu_i \in \mathbb{R}^3$ is uniformly sampled from $[0, 1]^3$, $\sigma_{ij}$ is uniformly sampled from $[0, 0.5]$, $p_i \in \mathbb{R}$ is uniformly sampled from $[0.2, 1]$ and normalized such that $\int_\Omega f(x)dx = 1$. Thus the source distribution $\mu$ is a complicated Gaussian mixture distribution restricted on $\Omega = [0, 1]^3$. After computing the OT map

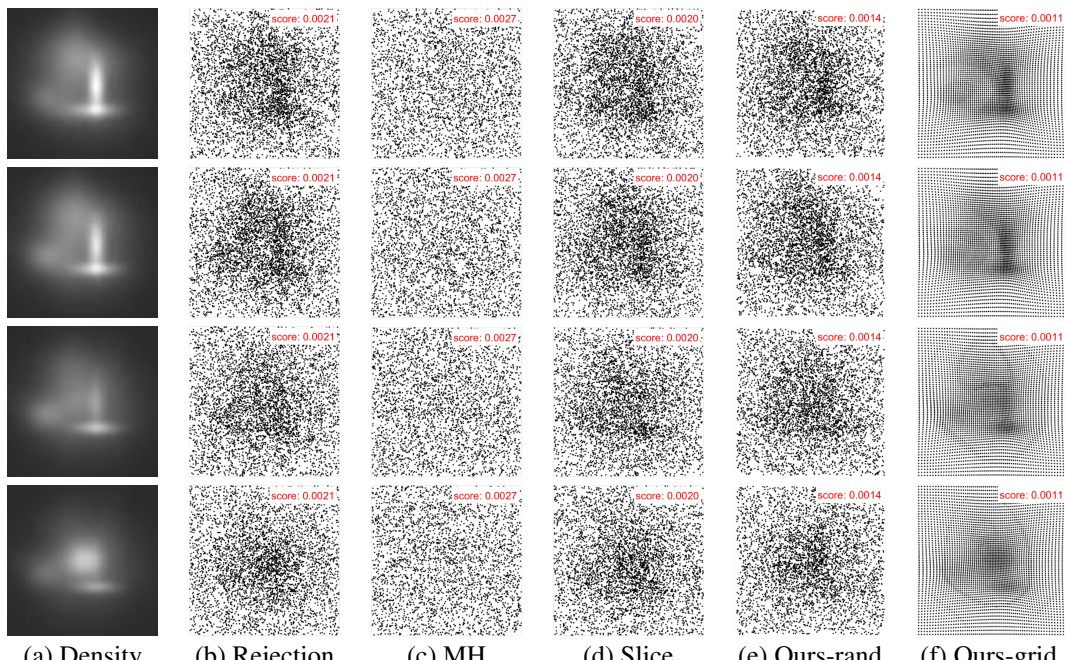

|     (a) Density     |     (b) Rejection     |     (c) MH     |     (d) Slice     |     (e) Ours-rand     |     (f) Ours-grid     |

**Figure 6:** 3D density function sampling. (a) The density functions in different slices of the same model, namely the 56th, 64th, 80th and 88th. (b)-(f) The samples obtained by different sampling methods. (b) Rejection sampling. (c) Metropolis-Hastings (MH) algorithm Bishop (2006). (d) Slice sampling Neal (2003). (e) The sampling results by mapping the random samples from the uniform distribution back to the desired distribution with $T^{-1}$. (f) The sampling results by mapping the grid centers back with $T^{-1}$. The scores of the top right give the results of the Chi-square goodness-of-fit test. Smaller means better. Zoom in for better visualization.

$T$ from $\mu$ to the uniform distribution $\nu$ defined on $[-1, 1]^3$, we conduct two groups of experiments: (i) we map the cell centers of the grid $\{y_k\}$ of $[-1, 1]^3$ back to $[-1, 1]^3$ through the inverse OT map $T^{-1}(y_k)$ defined by Eqn. (20); (ii) we randomly sample $100k$ samples $\{y_k\}$ from the Uniform distribution defined in $[-1, 1]^3$, then map them back to $[-1, 1]^3$ through the inverse OT map $T^{-1}(y_k)$. In order to keep the consistency with the mirror reflection process in the FFT-OT algorithm, we also reflect the the generated samples back to $\Omega$. To visualize the results of the $k$th slice, we plot the samples whose $z$ coordinates satisfy the inequality,

$$k/128 - 1/256 \leq z \leq k/128 + 1/256.$$

In Fig. 5 and Fig. 6, we give more sampling results of different slices correspond to the two models used in Fig. 2 in the main paper. Fig. 5 visualize the density function restricted on the 40th, 56th, 72th and 80th slices for different methods of the model displayed in the first row of 2. Fig. 6 visualize the density function restricted on the 56th, 64th, 80th and 88th slices for different methods of the model displayed in the second row of 2. Compared with the classical methods, the both sampling strategies of our method give decent sampling results that fit the prescribed density function well. Moreover, the number of generated samples for different slices of the same 3D model fits the density functions restricted to the corresponding slices well, namely more samples are generated in the brighter regions for different slices.

### D.2   MORE RESULTS ON VOLUMETRIC MAGNIFIER

In this experiment, we magnify the volumetric MRI image of the aneurysm by different amplification factors. In Fig. 7, we show the original aneurysm viewed from difference angles in the first column. The last three columns give the magnified results with different amplification factors from the viewpoints same as those in the first column. We can see that the aneurysm region is successfully magnified by different factors and the rest parts of the volume nearly keeps the same.

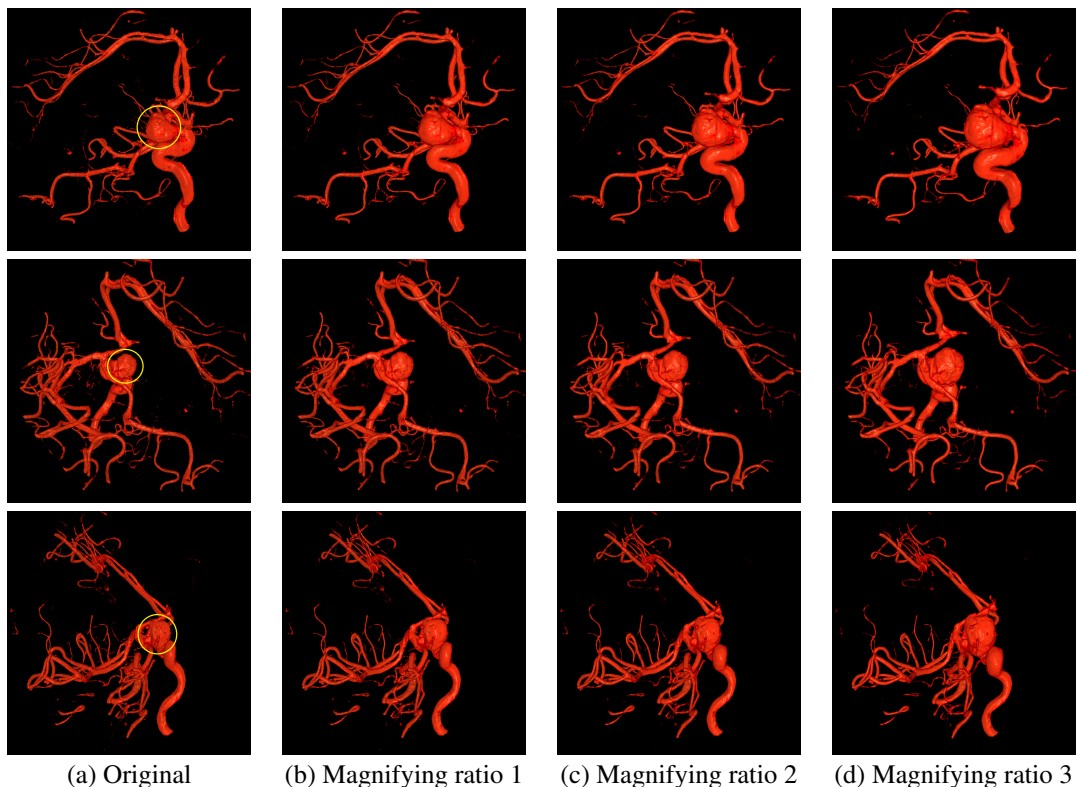

(a) Original      (b) Magnifying ratio 1      (c) Magnifying ratio 2      (d) Magnifying ratio 3

**Figure 7:** The volume magnifier of an aneurysm. The first column shows the original volumetric data from different viewpoints, and the last three columns give the magnified data from the same viewpoints of the first column with different magnifying ratios. The yellow circles denote the aneurysm or the ROIs.

