# OpenReview forum: "Volumetric Optimal Transportation by Fast Fourier Transform"
_ICLR.cc/2023/Conference — ICLR 2023 poster_

### Official Review · Reviewer_AQZc · 2022-10-24

**Confidence:** 3
**Correctness:** 3
**Technical Novelty And Significance:** 3
**Empirical Novelty And Significance:** 3
**Recommendation:** 6

**Clarity, Quality, Novelty And Reproducibility:**

The derivations are clear and the value of the work is clear, as optimal transport is a commonly arising problem in both machine learning and applied scientific computing. One minor note on clarity is that in some cases citations contain repeated author names: e.g. in the final paragraph of Section 1 "Though Lei and Gu [Lei & Gu] (2021) also uses FFT to solve the 2-dimensional OT problem..." and this occurs occasionally throughout.

One question that arose for me while reading was the use of the simplification in algorithm 1: g=1. This is not a valid simplification for optimal transport that I am aware of, could the authors provide some more reasoning for this simplification?

A central concern for me with this paper is novelty. As mentioned in the paper, Lei and Gu have already derived a very similar method. The algorithm is nearly identical, barring the fact that the aforementioned work only applies to 2d densities. I am not confident that this extension merits a full conference paper since the differences are mostly formal based on my understanding.

**Strength And Weaknesses:**

The main strength is that the method converges quickly and allows parallel calculation via gpus. It also provides a novel algorithm for nD approximate optimal transport (using the g=1, inversion trick).

A primary weakness in this work is that the method is not rigorously evaluated on relevant application areas for this conference. The most relevant experiments in Section 4.2, on adaptive sampling, are very limited. Only a single density is shown, and while the numerical comparisons presented to common sampling algorithms look good I don't think it can be said that this approach is much better for sampling. In part because of obvious patterns that appear in the transported grid centers. Perhaps this could be cleaned up with a more sophisticated algorithm, but then a more thorough analysis would be called for. I think the paper could benefit from that. The 3d magnification example is also lacking in my opinion, I cannot understand the need to perform 3d magnification with this method, as opposed to restricting a new viewing window to the ROI. For medical imaging, I suspect that the resulting distortion would make whatever benefits could be obtained through magnification useless.

**Summary Of The Paper:**

This paper develops a fast method for solving the optimal transport problem on 3d densities. The authors convert a linearized functional equation to a set of elliptic differential equations, and solve those using an FFT solver---converting finite differences scheme to an algebraic system by passing to the frequency space. The authors perform a convergence analysis and show applications that highlight the usefulness of the method.

**Summary Of The Review:**

Contributions in this field are important for a wide range of machine learning problems. I think that the result in this paper are promising and worthy of publication in some forum. However, the experimental analysis and is not in a condition to be published presently. More extensive analysis on the use of 3d FFT-OT for sampling could result in a more appropriate submission in my opinion, including comparison of runtime and more thorough empirical and theoretical analysis of the algorithm used as a sampler.

---
I have updated more overall assessment of the contribution after more carefully reviewing the literature. The authors addressed my concerns with the g=1 assumption, which I believe that they should make more explicit in their manuscript. Other than that I think the work is acceptable, if slightly below my standards for rigorous empirical analysis. Time permitting, experiments assessing the quality of the sampling algorithm they propose on nD canonical distributions (e.g. Cauchy, Gaussian, comb) from the standpoint of information theoretic metrics would overcome my reservations here, but I don't think they're strictly necessary for publication.

---

> ### Author Response · Authors · 2022-11-19
> **Thanks for the review**
>
> We thank the reviewer AQZc for the reviews, and we have revised the manuscript accordingly.
>
> > A primary weakness in this work is that the method is not applied to any particularly relevant areas for this conference.
>
> Please check the 'Subject Areas' of ICLR 2023, one of them is **optimal transport**. Therefore, our paper definitely aligns with the conference.
>
> > The most relevant experiments in Section 4.2, on adaptive sampling, are very limited. Only a single density is shown.
>
> In Fig. 2, we give **two** density sampling examples, **not one**. The slices of the first and second rows come from two different 3D density functions, which are from complex Gaussian mixture models with 30 components defined in Sec. 4.1. To show the performance of our method, we manually pick up the slices with clear patterns. More slices can be found in Fig. 5 and Fig. 6 in the Appendix, which can help illustrate the complexity of the density functions and the performance of the proposed method.
>
> > While the numerical comparisons presented to common sampling algorithms look good I don't think it can be said that this approach is much better for sampling. In part because of obvious patterns that appear in the transported grid centers. Perhaps this could be cleaned up with a more sophisticated algorithm, but then a more thorough analysis would be called for. I think the paper could benefit from that.
>
> - We not only visually compare with the classical methods in Fig.2, Fig. 5 and Fig. 6. To quantify the comparison, we also use the Chi-square goodness-of-fit test to evaluate the sample quality. The scores given in the top-right corner of each sampling method in the figures show that our proposed method achieves better results than the compared methods.
> - **Though both qualitatively and quantitatively, we show that our method beat the classical methods**, we'll appreciate it if the reviewer can provide some better sampling or analysis methods.
>
> > The 3d magnification example is also lacking in my opinion, I cannot understand the need to perform 3d magnification with this method, as opposed to restricting a new viewing window to the ROI.
>
> - With our method, the users can use mouse to click a specific point or region, the the ROI will be magnified accordingly. In such a way, the doctors can easily examine different ROIs with the whole subject in the field of view. This should help them grasp the detailed local information and the global information at the same time.
> - If using a new window of the ROI as stated by the reviewer, the doctors need to view different windows (one for the whole subject and one for the ROI) back and force, which is tedious and inefficient. It is also hard for the doctors to grasp the relationship between the ROI and the whole subject with two windows. Actually, this obstacle motivated us to propose the OT based 3D magnifier.
>
> > The use of the simplification in algorithm 1: g=1. This is not a valid simplification for optimal transport that I am aware of, could the authors provide some more reasoning for this simplification?
>
> - By setting $g=1$, we can make the solving of the Monge-Ampere equation tractable, and the solution itself can give many real applications like those in Sec. 4.
> - Similar to [1], our method can be easily extended to obtain the transport map between any two density functions $d\mu = f(x)dx$ and $d\nu = g(y)dy$. Firstly, we can compute the OT map $T_1$ from $\mu$ to the Uniform distribution and the OT map $T_2$ from $\nu$ to the same Uniform distribution. Then $T=T_1^{-1}\circ T_2$ will give us an effective transport map from $\mu$ to $\nu$. Also, equation (20) provides the method to compute the inverse map. Though the map $T$ may not be optimal, Nader and Guennebaud [1] show that it is a good approximation of the OT map.
>
> > A central concern for me with this paper is novelty. As mentioned in the paper, Lei and Gu have already derived a very similar method. The algorithm is nearly identical, barring the fact that the aforementioned work only applies to 2d densities. I am not confident that this extension merits a full conference paper since the differences are mostly formal based on my understanding.
>
> As we stated in the last paragraph of Sec. 1, though Lei and Gu also use FFT to solve the **2-dimensional** OT problem, our method is different form their work in the following three aspects:
> - their method uses the **fixed point method to compute the 2D OT problems**, ours is based on **the linearization of the Monge-Ampere operator to solve the 3D OT problems**, these are two different methodologies in PDE theory;
> - their method cannot be extended to solve 3D problems while ours can;
> - in our paper, we also provide the theoretical convergence analysis of the proposed method. In Lei and Gu's paper, no theoretical convergence is guaranteed.
>
> **Reference**
>
> [1] Georges Nader and Gael Guennebaud, Instant Transport Maps on 2D Grids, ACM Trans. Graph., 2018.

---

> > ### Comment · Reviewer_AQZc · 2022-11-23
> > **Partial response**
> >
> > > Please check the 'Subject Areas' of ICLR 2023, one of them is optimal transport. Therefore, our paper definitely aligns with the conference.
> >
> > I fully agree, I verified this prior to review. This comment was just to indicate that the applications do not emphasize the relationship to the "learning of representations". The sampling example has the potential to do so but I think it comes up short.
> >
> > > In Fig. 2, we give two density sampling examples, not one. The slices of the first and second rows come from two different 3D density functions, which are from complex Gaussian mixture models with 30 components defined in Sec. 4.1. To show the performance of our method, we manually pick up the slices with clear patterns. More slices can be found in Fig. 5 and Fig. 6 in the Appendix, which can help illustrate the complexity of the density functions and the performance of the proposed method.
> >
> > Several anecdotal exampled (contrives ones at that [why use these apparently random mixtures and not a fixed density?]) do not show the usefulness of this method for the problem, they simply show the capability of the approach. I do not think that this approach must produce state of the art results for publication, but a more detailed empirical investigation of the nature of this 3d transport algorithm for sampling would improve the papers standing. I have used sampling approaches like these in several previous applications and I can't understand why I should use your method as opposed to more conventional methods that are simpler.
> >
> > > If using a new window of the ROI as stated by the reviewer, the doctors need to view different windows (one for the whole subject and one for the ROI) back and force, which is tedious and inefficient. It is also hard for the doctors to grasp the relationship between the ROI and the whole subject with two windows. Actually, this obstacle motivated us to propose the OT based 3D magnifier.
> >
> > Can you provide a specific citation for the inefficiency of this? Is this specifically mentioned in the citations in the work? This sounds more like a UI problem with a contrived solution that an actual problem for me, but I admit I am not an expert in medical imaging user interfaces and would yield to published evidence of the shortcomings of conventional viewing interfaces for (eg.) diagnostics.
> >
> > Actually, this criticism of the empirical evidence supporting the work being superficial (in my opinion) applies to both applications. In the graphics literature I think that there are often applications of optimal transport to mesh or shape registration. Some publications in the machine learning literature [https://www.jmlr.org/papers/volume20/18-079/18-079.pdf?ref=https://githubhelp.com] show much more extensive empirical evaluation. I agree that a very significant theoretical contribution can override empirical shortcomings, but why not perform these experiments to better justify to this community?
> >
> > I am spending some more time to understand your rebuttal to my novelty and the g=1 concerns. I will reply with my thoughts on these soon. I think these are the key issues for really understanding the impact of this work since my concerns about the empirical side remain and I frankly need a little more time thinking about this.

---

> > > ### Author Response · Authors · 2022-11-26
> > > **Authors' response for the updated comment**
> > >
> > > We thank the reviewer for the comments. Here we want to emphasize that our main contribution is that we propose the most powerful algorithm for the 3D continuous OT problem. If possible, the reviewer may give some comments about the method itself. It seems unfair to reject a paper merely based on the reviewer's personal thinking about the applications.
> > >
> > > > The applications do not emphasize the relationship to the "learning of representations".
> > >
> > > When making the submission, we didn't mention anything about 'learning of representation'. It's a paper about 'general machine learning'.
> > >
> > > > I do not think that this approach must produce state of the art results for publication, but a more detailed empirical investigation of the nature of this 3d transport algorithm for sampling would improve the papers standing.
> > >
> > > As we stated in Sec 4.2 and Fog. 2, we have already compared with bunch of methods, including rejection sampling, Metropolis-Hastings (MH) algorithm and Slice sampling, and experimental results show that our method achieves the best performance both qualitatively and quantitatively. If the reviewer does think that we need to do more comparison, is it possible to provide some references or names?
> > >
> > > > I have used sampling approaches like these in several previous applications and I can't understand why I should use your method as opposed to more conventional methods that are simpler.
> > >
> > > Our method can provide better sampling quality. If you need high quality samples, you can try our method.
> > >
> > > > Can you provide a specific citation for the inefficiency of this? Is this specifically mentioned in the citations in the work? This sounds more like a UI problem with a contrived solution that an actual problem for me, but I admit I am not an expert in medical imaging user interfaces and would yield to published evidence of the shortcomings of conventional viewing interfaces for (eg.) diagnostics.
> > >
> > > - To the best of our knowledge, we are the first that propose the idea of 3D magnifier, and only through 3D optimal transport, we can achieve this. So there are no references or citations.
> > > - We did consult some doctors about the application of 3D magnifier in medical imaging, and got the conclusion that it has the potential to help the doctors in the future.
> > >
> > > > In the graphics literature I think that there are often applications of optimal transport to mesh or shape registration. Some publications in the machine learning literature [https://www.jmlr.org/papers/volume20/18-079/18-079.pdf?ref=https://githubhelp.com] show much more extensive empirical evaluation.
> > >
> > > - For the paper the reviewer provided, it aims to solve **discrete OT** problem, while ours try to solve **continuous OT** problem, they are two directions in OT area.
> > > - The provided reference shows that by using a subset of the full dataset to compute the discrete OT, their results can approach the ones with the full dataset. If the reviewer here means the convergence of the algorithm, please see Theorem 3 of the main paper, where we prove that our method will converge to the ground-truth linearly.
> > > - We agree that there are many other applications of 3D OT, like shape registration. In this paper, we have already used the 3D adaptive sampling and 3D magnifier to help illustrate the potential usage of our proposed algorithm. We don't think it is necessary to cover all of the applications. After all, here our most important contribution is to propose the most efficient 3D OT algorithm.

---

> > > > ### Comment · Reviewer_AQZc · 2022-11-29
> > > > **Continued response**
> > > >
> > > > After carefully reviewing the literature I feel I have a better understanding of the theoretical import of the work. My concerns about redundancy or over-similarity with existing works are not substantial, so I think that I can improve my assessment of the novelty and contribution, e.g. [Lei+Gu, '21]. [Lei+Gu, '21] introduced the FFT-OT algorithm for solving 2d optimal transport problems and this introduction is inherently more novel to me than n-D extensions. However, there is clearly value in the higher dimensional extensions and the algorithmic solution depends on a different line of theoretical rasoning. I admit that I was overly harsh in my assessment, especially after more carefully understanding the literature.
> > > >
> > > > I still find the empirical evaluation weak. I think that the paper would significantly benefit from more extensive and quantitative evaluation of the value of the method for the applications proposed. This is for 2 reasons: 1) the result is an approximation as many optimal transport algorithms are, and so the biases in the algorithm may have impacts on the proposed application areas, 2) the algorithm claims to extend to generic n-D problems yet only evaluates on 3D---as the other reviewer mentioned the storage capacity scaling may limit practicality for higher dimensional applications. Perhaps this can be left for subsequent work, but I think there is still work to do to really understand the applied impact of this paper (and in this field OT is a methodology for alignment hence application feasibility is important).
> > > >
> > > > Please find my review updated.
> > > >
> > > > ----
> > > >
> > > > Specific responses:
> > > > > As we stated in Sec 4.2 and Fog. 2, we have already compared with bunch of methods, including rejection sampling, Metropolis-Hastings (MH) algorithm and Slice sampling, and experimental results show that our method achieves the best performance both qualitatively and quantitatively. If the reviewer does think that we need to do more comparison, is it possible to provide some references or names?
> > > >
> > > > The paper which you looked at and discussed below has this much more extensive evaluation for the density estimation experiments.
> > > >
> > > > > [For the paper the reviewer provided [https://www.jmlr.org/papers/volume20/18-079/18-079.pdf], it aims to solve discrete OT problem, while ours try to solve continuous OT problem, they are two directions in OT area.]
> > > >
> > > > This is not my point, the point was that they did a more extensive experimental evaluation in their work at a peer publication (JMLR, granted this is a journal). BTW they mentioned (of course) that provided you can sample from the continuous densities, their approach can solve continuous transport approximately---since your method depends on FT coefficients you would also need to have samples or a closed-form FT. [Lei + Gu, '21] also did a more extensive experimental evaluation than your work, however I also find their evaluation to be anecdotal. At least in their case (and in 2d) we can get a better sense of the effects of using the method for the proposed (UV mapping) application. This is why I mentioned 3d shape registration: volumetric registration could be done with your method.
> > > >
> > > > About the g=1 assumption: I think that you should outline the limitations of this more explicitly in your work. The title "Optimal transport" is a bit misleading because in reality this algorithm can not do optimal transport between arbitrary densities, but just down to a uniform density and use the composed inverse with target to solve for an approximate optimal transport.

---

> > > > > ### Author Response · Authors · 2022-12-07
> > > > > **Thanks for the updated reviews and acknowledgement of our work**
> > > > >
> > > > > We thank the reviewer for your detailed reviews and the acknowledgement of our work and rebuttal. Please see the updated answers to the follow-up questions.
> > > > >
> > > > > > More extensive experimental evaluation like volumetric registration.
> > > > >
> > > > > We agree that the experiments about volumetric shape registration can help better illustrate the proposed method. We are designing experiments about the registration of different volumetric MRI data obtained in different time for the same patient. In such a way, the proposed method can automatically find any potential changes. Due to the limited time, we may not make it before the deadline of the rebuttal, so we'll leave this part for the future work.
> > > > >
> > > > > > Clarify the $g=1$ assumption in the manuscript.
> > > > >
> > > > > We have added this point to the manuscript as a limitation of the proposed method in Sec. 1, the details are as follows:
> > > > >
> > > > > *(iii) In this work, we mainly focus on the computation of the OT map from the uniform distribution to another arbitrary continuous distribution. To extend the method to find the OT map between any two continuous measures, we can compute two OT maps from the uniform distribution to the both continuous measures, then combine them together. The combination will give a reasonable approximation of the OT map \cite{Nader2018MAOT}.*

---

### Official Review · Reviewer_wDkb · 2022-10-25

**Confidence:** 4
**Correctness:** 4
**Technical Novelty And Significance:** 4
**Empirical Novelty And Significance:** 3
**Recommendation:** 8

**Clarity, Quality, Novelty And Reproducibility:**

The paper is clear and novel.

The figures need to be improved by providing zoom factors and highlighting both upper and lower rows.

The code is not provided. Therefore the paper is hard to reproduce. However, the major algorithms are provided and they seem straightforward.

**Strength And Weaknesses:**

The methods is very fast but needs more memory (storage complexity increases exponentially). The method can be extended to work with any number of dimensions.

The connection between volumetric magnification and denoising or super resolution should be clarified.

Figures 3 and 4 do not show the magnification ratios used.

The following statement needs clarification :
"Compared with the traditional method requiring tedious zoom in/out, our method only magnifies the
ROI region and keeps the whole subject in the field of view, which enables doctors to visualize the
overall anatomy while scrutinize detailed anatomical structure at the same time."

Not clear why this is better than magnifying the entire volume. Isn't it puzzling for the doctor looking at some regions being zoomed?


**Summary Of The Paper:**

The authors propose a new method based on Fast Fourier to compute 3D optimal transport problems.

The authors show great speedups and provide theoretical insights using elliptic PDEs. Applications in sampling and for magnification of volumetric data.

**Summary Of The Review:**

Overall, this is novel work and valuable. The speedups and theoretical connections are impressive.

---

> ### Author Response · Authors · 2022-11-19
> **Thank you for the review and support**
>
> We thank the reviewer wDkb for the through reviews, and we have revised the manuscript accordingly.
>
> > The connection between volumetric magnification and denoising or super resolution should be clarified.
>
> - OT based volumetric magnification can be treated as a sophisticated local liner interpolation method to **magnify the geometry of ROI**, as explained in equation (20). Then by applying **3D texture mapping**, we finally get the magnified volumetric data.
> - For denoising or super resolution methods, they are usually **global methods directly applied to the 3D textures**. Furthermore, to obtain high quality 3D denosing or super resolution results, we usually need supervised models trained with large scale dataset, which is hard to obtain for 3D volumetric data.
>
> > Figures 3 and 4 do not show the magnification ratios used.
>
> - To obtain the results of the last 3 columns of Fig. 3, we set $\sigma=\sigma_x = \sigma_y = \sigma_z$, and they are 0.83, 0.75 and 0.5 respectively.
> - For Fig.4, we set $\sigma_x = \sigma_y = \sigma_z = 0.75$ for all of the 4 experiments.
>
> > Not clear why this is better than magnifying the entire volume. Isn't it puzzling for the doctor looking at some regions being zoomed?
>
> In our design, the 3D magnifier is similar to 2D magnifier. The users can use mouse to click a specific point or region, the the ROI will be magnified accordingly. In such a way, the doctors can easily examine different ROIs with the whole subject in the field of view. This should help them grasp the detailed local information and the global information at the same time, instead of tedious zooming in/out through different regions.
>
> > Code
>
> We'll open source the code upon the paper is accepted. Basically, the algorithm is implemented by C++ with Cuda.

---

### Official Review · Reviewer_Q68w · 2022-11-03

**Confidence:** 4
**Correctness:** 3
**Technical Novelty And Significance:** 3
**Empirical Novelty And Significance:** Not applicable
**Recommendation:** 6

**Clarity, Quality, Novelty And Reproducibility:**

# Clarity
The submission is relatively clear, but has a large number of typos and minor inconsistencies. We list the main typos and inconsistencies here.

1. The role of $\tau$ as described in [1] is as a stabilization factor, not a step length as stated in Algorithm 1. Furthermore, I believe the update of the Brenier potential in the same algorithm should be $u_n+v_n$, not $u_n+\tau v_n$ to match with the reference [1] (also $\tau$ should in practice be large).
2. The scores in Figure 5 and Figure 6 do not vary over the rows. This seems like a mistake, as the scores in Figure 2 do vary.
3. In Appendix B.2. and B.3. many of the norms should be squared, but are not.
4. There is some notation in the proof of the main theorem which is not defined, see the weaknesses section for a more in depth discussion.
5. Names with accents are spelled inconsistently. The correct spellings are Monge-Ampère and Poincaré. This does not effect clarity, but should be fixed.

# Quality

The paper includes a number of technical errors as discussed in the weaknesses section.

# Novelty

The novelty of this work is reasonable. Although it presents only a simple modification to the work of [1] and [2] via averaging of the coefficients of the variable coefficient PDEs and applying a spectral method rather than solving these variable coefficient PDEs directly as in [2], the simplicity and speed of the proposed method is certainly of interest.


# Reproducibility

The main algorithmic details of the paper would be simple to reproduce. My only concern is the lack of discussion for how $\tau$ is chosen so as to guarantee linear rates of convergence (see the Weaknesses section for additional details).

## References

[1]: G. Loeper and F. Rapetti, Numerical solution of the Monge-Ampère equation by a Newton's algorithm. _C. R. Acad. Sci. Paris_, Ser. I 340(4), 2005.

[2]: L.-P. Saumier, M. Agueh, and B. Khouider, An efficient numerical algorithm for the L2 optimal transport problem with periodic densities. _IMA J. Appl. Math._, 80(1), 2015.

**Strength And Weaknesses:**

# Strengths
- This submission builds nicely upon the previous work of Loeper-Rapetti [1] and Saumier et al. [2] by providing a fast and parallelizable spectral approach to solving the Monge-Ampère equation characterizing the OT map in this particular setting. This algorithmic improvement is enabled by the simple approach of averaging the coefficients of the variable coefficient PDEs obtained from the linearization approach discussed in [1] and [2].

- The applications to density sampling and volume magnification are of interest and the proposed method appears to give good results in the provided experiments.

# Weaknesses

 - In contrast to [2], the present submission does not cover the case where the target distribution is not the Lebesgue measure. The source distribution is also required to satisfy some conditions which restrict the general applicability of this method.

-Theorem 3 states that "one can choose the step length parameter $\tau$ such that [...] the algorithm has a linear convergence rate." However, per the proof of this Theorem, the choice of $\tau$ depends on the uniform ellipticity properties of the solution to the problem. There is, in particular, no discussion on how $\tau$ was chosen in the experiments, nor how one should choose $\tau$ in general.

-The submission also contains a number of technical errors which we list below
1. The entire paper appears to work under the assumption that the Brenier potential $u\in C^2(\Omega)$. As discussed in Theorem 2.1 in [2], this requires some additional conditions on the density of $\mu$ which are not mentioned in the present submission.

2. In equation (26), it is claimed that the integral over the boundary is equal to 0 as $\partial \Omega$ has zero measure. The reasoning here is erroneous, as although $\partial \Omega$ has zero Lebesgue measure, integrals over the boundary are taken with respect to the relevant surface measure, not the Lebesgue measure and $\partial \Omega$ has positive measure with respect to the surface measure.

3. Although it is not explicitly mentioned in the submission, the density of $\mu$ with respect to the Lebesgue measure must be positive on $\Omega$ for some of the claims to hold.

4. In the introduction, the OT map is stated to define a distance between any two probability distributions. This is incorrect, in particular there exists no transport map between a Dirac mass supported on one point and a normalized sum of two Dirac masses. It is rather the Kantorovich problem and corresponding OT couplings which are used to define such a distance.

5. Problem 1 is stated in terms of measure preserving maps which, in turn, are only defined in this manuscript in the context of $C^1$ maps. As such the statement of Brenier's theorem provided (Theorem 2) is incorrect, as the OT map provided by the true Brenier theorem need not be $C^1$, rather it is only guaranteed to be differentiable $\mu$-a.e. by Alexandrov's theorem under the present assumptions (cf. eg. Theorem 3.1 and Theorem 3.2 in [3]). Similarly, Problem 1 is typically stated as an optimization problem over arbitrary measurable maps satisfying the pushforward condition $T_{\sharp} \mu =\nu$ i.e. $\mu(T^{-1}(A))=\nu(A)$ for every Borel set $A$.

6. The proof of the main theorem is difficult to follow to the point where I cannot guarantee its accuracy. In particular, the notation $a_k, A_k,\bar A_k$ is undefined despite these objects playing a central role in the proof. Moreover, I do not understand the meaning of the display after equation (37),
$(f_h-\rho_{h,k})^TA_k\bar A_k^{-1}(f_h-\rho_{h,k})=\text{tr} ((f_h-\rho_{h,k})^TA_k\bar A_k^{-1}(f_h-\rho_{h,k}))$, it is obvious that the trace of a scalar is equal to itself.

## References

[1]: G. Loeper and F. Rapetti. Numerical solution of the Monge-Ampère equation by a Newton's algorithm. _C. R. Acad. Sci. Paris_, Ser. I 340(4), 2005.

[2]: L.-P. Saumier, M. Agueh, and B. Khouider. An efficient numerical algorithm for the L2 optimal transport problem with periodic densities. _IMA J. Appl. Math._, 80(1), 2015.

[3]: G. De Philippis and A. Figalli. The Monge–Ampère equation and its link to optimal transportation. _Bull. Am. Math. Soc_, 51(4), 2014.

**Summary Of The Paper:**

This submission proposes an algorithm for numerically estimating the optimal transport (OT) map in dimension $3$ between a source distribution $\mu$ admitting a (Hölder) continuous density on $[0,1]^3$, and the Lebesgue measure on $[0,1]^3$. In this context, the optimal transport map can be characterized as the solution of a nonlinear degenerate elliptic partial differential equation (PDE) known as the Monge-Ampère equation.

The proposed method consists of applying the classical method of continuity to obtain a one-parameter family of PDEs $(\det(D_x^2u(x,t))=\rho(x,t))_{t\in[0,1]}$ for which $u(x,0)$ is known and $u(x,1)$ can be used to obtain the OT map. Next, this family is discretized with respect to its parameter, linearized, and the resulting variable coefficient linear elliptic PDEs are made into constant coefficient PDEs by averaging the coefficients. Given the solution at the previous step, it is updated by solving the resulting constant coefficient PDE; which is amenable to computation by a fast and parallelizable spectral method based on the Fast Fourier Transform (FFT) implemented on GPUs.

The algorithm is shown to benefit from a linear rate of convergence. Applications to density sampling and volume magnification are also included.

**Summary Of The Review:**

The method proposed in this submission is of interest due to its simplicity and speed. The main novelty is to average the coefficients of the linearized PDEs to enable a spectral approach. The proposed applications of this work are also of interest. As it stands, however, the submission has a number of technical mistakes and thus I cannot recommend acceptance of this work in its current state.

## Post Rebuttal Comments:

In their rebuttal, the authors have fixed many of the technical issues that were brought up in the previous review. The scores have been amended to reflect these changes. Please also address the following list of issues; some of which were mentioned in the original review, but were not fixed in the revised manuscript. I also suggest that the revised manuscript be proofread to fix typos.

1. Spelling of names with accents are still inconsistent.
2. In the appendix many of the norms should be squared, but are not.
3. After equation (37), the sentence "The scalar" followed by the displayed equation does not make sense. I would recommend replacing that sentence by "Remark that the following displayed equation is a scalar" if this emphasis is deemed to be necessary.
4. $o(\tau^{-1})$ is simply dropped after equation (37) without further comment, this is somewhat sloppy.
5. In the display before (38), the upper bound is not used in the remainder of the proof to my understanding; if that is the case, the upper bound can just be dropped. The lower bound in the same display could use a bit more explanation to make it more clear to the reader.
6. It would be useful to add a comment on why $\bar{A}_k$ is invertible before (37).

---

> ### Author Response · Authors · 2022-11-19
> **Thanks for the thorough and detailed review**
>
> We thank the reviewer Q68w for the valuable comments, which are crucial for us to improve the rigor of our manuscript. The main concern is about the regularity of the OT map, so in the manuscript we add the following paragraph to address this question:
>
> The existence, uniqueness and regularity of OT maps depend on the boundedness and the continuity of the density functions, the convexity of the supporting domains, the continuity of their boundaries, and the cost functions. In our current work, we focus on the similar situation in [2],
> - The cost function is quadratic Euclidean distance $c(x,y)=\|x-y\|^2/2$;
> - The supports of the source and the target measures are the canonical cube $\Omega = [-1,1]^3$, which is uniformly convex;
> - The source and the target measures $\mu,\nu$ are absolutely continuous with respect to the Lebesgue measure, their densities $f,g$ are positive and bounded away from zero;
>     $0 < m <f,g < M$
>     and $f,g$ are of class $C^\alpha(\Omega)$,
> - The boundary condition is second boundary condition (OT boundary condition), $T(\Omega)=\Omega$.
> Then according to (Villani Theorem 14.4, Theorem 2.1 in [2]), the OT map $T:\Omega\to \Omega$ exists and is unique and invertible ($\mu$ a.e), and the Brenier potential is of class $C^{2,\beta}(\Omega)$ for some $0<\beta < \alpha$.
>
>
> > In contrast to [2], the present submission does not cover the case where the target distribution is not the Lebesgue measure. The source distribution is also required to satisfy some conditions which restrict the general applicability of this method.
>
> The work in [2] also proposed to use FFT to solve Monge-Ampere equation for optimal transport maps. The work in [2] focuses on periodic boundary condition, but our proposed work focuses on general second boundary condition; the work in [2] concerns planar OT maps, ours emphasizes on volumetric OT maps, which has higher complexity.
>
>
> > Theorem 3 states that "one can choose the step length parameter $\tau$  such that [...] the algorithm has a linear convergence rate." However, per the proof of this Theorem, the choice of $\tau$ depends on the uniform ellipticity properties of the solution to the problem. There is, in particular, no discussion on how  was chosen in the experiments, nor how one should choose  in general.
>
> Similar to the damped Newton method in [1] and [2], the step length is crucial for the convergence. In our experiments, we initially choose a large step length $\tau$ and if the algorithm collapse, we shrink $\tau$ to half and try again. According to the theorem 3.1, if $f\in C^{2,\alpha}(\Omega)$ and $g\in C^{3,\alpha}(\Omega)$, there exists a $\tau>0$ which guarantees the convergence.
>
>
> > The entire paper appears to work under the assumption that the Brenier potential $u\in C^2(\Omega)$. As discussed in Theorem 2.1 in [2], this requires some additional conditions on the density of $\mu$ which are not mentioned in the present submission.
>
> We add the clarification that the source density is of class $C^\alpha(\Omega)$, which guarantees the Brenier potential to be $u\in C^2(\Omega)$.
>
> > In equation (26), it is claimed that the integral over the boundary is equal to 0 as $\partial\Omega$ has zero measure. The reasoning here is erroneous, as although $\partial\Omega$ has zero Lebesgue measure, integrals over the boundary are taken with respect to the relevant surface measure, not the Lebesgue measure and $\partial\Omega$ has positive measure with respect to the surface measure.
>
> We have corrected the claim as: **The integral over the boundary is equal to $0$ because $v$ satisfies the Neumann boundary condition.**
>
> > Although it is not explicitly mentioned in the submission, the density of $\Omega$ with respect to the Lebesgue measure must be positive on  for some of the claims to hold.
>
> In our algorithm 1, we require $f>0$. We have explicitly added this to the manuscript.
>
> > In the introduction, the OT map is stated to define a distance between any two probability distributions. This is incorrect, in particular there exists no transport map between a Dirac mass supported on one point and a normalized sum of two Dirac masses. It is rather the Kantorovich problem and corresponding OT couplings which are used to define such a distance.
>
> We have changed the sentence to **the optimal transport plan defines a distance between them. If $\mu$ and $\nu$ satisfies some general conditions, and the transport cost satisfies the twisting condition, then the optimal transport plan becomes the optimal transport map.**

---

> > ### Author Response · Authors · 2022-11-19
> > **Continue**
> >
> > > Problem 1 is stated in terms of measure preserving maps which, in turn, are only defined in this manuscript in the context of $C^1$ maps. As such the statement of Brenier's theorem provided (Theorem 2) is incorrect, as the OT map provided by the true Brenier theorem need not be $C^1$, rather it is only guaranteed to be differentiable $\mu$-a.e. by Alexandrov's theorem under the present assumptions (cf. eg. Theorem 3.1 and Theorem 3.2 in [3]). Similarly, Problem 1 is typically stated as an optimization problem over arbitrary measurable maps satisfying the pushforward condition $T_#\mu = \nu$ i.e.  for every Borel set $A$.
> >
> > - We have changed the definition of measure-preserving map to the general form, and modified theorem 2 as follows:
> > **Suppose $T:\Omega\to \Sigma$ is a measurable map. The mapping $T$ is called \emph{measure preserving} and denoted as $T_#\mu = \nu$ if the following relation $\mu(T^{-1}(A))=\nu(A)$
> > for any every subset $A\subset \Sigma$.**
> > - Theorem 2 assumes that $\Omega$, $\mu,\nu,f$ and $g$ are defined as above. Then there exists a convex function $u:\Omega\to\mathbb{R}$, $u\in C^{2,\beta}(\Omega)$ for some $0<\beta<\alpha$,
> > such that $\nabla u$ pushes $\mu$ forward to $\nu$, $(\nabla u)_{\#}\mu=\nu$.
> >
> > &nbsp;&nbsp;&nbsp;&nbsp;&nbsp;&nbsp; Moreover, $\nabla u$ is unique and invertible ($\mu$ - a.e), and its inverse $\nabla v$ satisfies $(\nabla v)_{\#}\nu=\mu$.
> >
> > > The proof of the main theorem is difficult to follow to the point where I cannot guarantee its accuracy. In particular, the notation $a_k, A_k, \bar{A}_k$ is undefined despite these objects playing a central role in the proof. Moreover, I do not understand the meaning of the display after equation (37), it is obvious that the trace of a scalar is equal to itself.
> >
> > - In the $k$-th step, we solve an elliptic PDE with variant coefficient, the bilinear form is defined as $a_k$ in Eqn.(27). By using the Galerkin's method, the bilinar form Eqn.(29) is represented as a stiffness matrix $A_k$ in the left hand side of Eqn. (30), which varies from point to point. By taking the average, we obtain the mean stiffness matrix $\bar{A}_k$ (as the process from Eqn.(10) to  Eqn. (11) ).
> > - The display after equation （37）is to emphasize the left hand side is a scalar.
> >
> > **Reference**
> >
> > [1] G. Loeper and F. Rapetti. Numerical solution of the Monge-Ampère equation by a Newton's algorithm. C. R. Acad. Sci. Paris, Ser. I 340(4), 2005.
> >
> > [2] L.-P. Saumier, M. Agueh, and B. Khouider. An efficient numerical algorithm for the L2 optimal transport problem with periodic densities. IMA J. Appl. Math., 80(1), 2015.
> >
> > [3] G. De Philippis and A. Figalli. The Monge–Ampère equation and its link to optimal transportation. Bull. Am. Math. Soc, 51(4), 2014.
> >
> > [4] Georges Nader and Gael Guennebaud, Instant Transport Maps on 2D Grids, ACM Trans. Graph., 2018.

---

> > > ### Comment · Reviewer_Q68w · 2022-11-29
> > > **Response to Rebuttal**
> > >
> > > These comments and the corresponding updates to the manuscript have addressed the majority of the concerns that were brought up in my original review. I have added a few more comments to the review after the "Summary Of The Review" section under the heading "Post Rebuttal Comments" and have updated the score correspondingly. If these final issues can be addressed I believe the submission will be of reasonable quality.

---

> > > > ### Author Response · Authors · 2022-12-07
> > > > **Updated rebuttal**
> > > >
> > > > We thank the reviewer for your detailed reviews and the acknowledgement of our work and rebuttal. Please see the updated answers to the post rebuttal comments.
> > > >
> > > > > Spelling of names with accents are still inconsistent.
> > > >
> > > > We have modified all the names consistently as Amp\`ere and Poincar\'e. We'll also continue to check the manuscript and fix any typos.
> > > >
> > > > > In the appendix many of the norms should be squared, but are not.
> > > >
> > > > We have followed the advice and corrected the norms to be squared.
> > > >
> > > > > After equation (37), the sentence "The scalar" followed by the displayed equation does not make sense. I would recommend replacing that sentence by "Remark that the following displayed equation is a scalar" if this emphasis is deemed to be necessary.
> > > >
> > > > We have followed the advice and replace "The scalar'' by "Remark that the following displayed equation is a scalar''.
> > > >
> > > > > $\sigma(\tau^{-1})$ is simply dropped after equation (37) without further comment, this is somewhat sloppy.
> > > >
> > > > We have added the following sentence ``In the following discussion, the term $o(\tau^{-1})$ will be ignored.''
> > > >
> > > > > In the display before (38), the upper bound is not used in the remainder of the proof to my understanding; if that is the case, the upper bound can just be dropped. The lower bound in the same display could use a bit more explanation to make it more clear to the reader.
> > > >
> > > > We have followed the advice to drop the upper bound, and only focus on the lower bound.
> > > >
> > > > > It would be useful to add a comment on why $\bar{A_k}$ is inevitable before (37).
> > > >
> > > > By the  uniform ellipticity Eqn. (22), the eigen values of the adjoint matrix $(u^{pq})(x,t)$ is uniformly bounded away from zero in the space $\mathcal{H}:=\{\xi\in \mathbb{R}^n|\sum_i \xi_i=0\}$, so the eigen value of the mean adjoint matrix $\bar{u}^{pq}(t)$ is bounded away from zero in $\mathcal{H}$. After discretization, the eigen values of $\bar{A}_k$ is strictly positive in $\mathcal{H}$, hence $\bar{A}_k$ is invertible in $\mathcal{H}$.
> > > >
> > > >
> > > > > The scores in Figure 5 and Figure 6 do not vary over the rows. This seems like a mistake, as the scores in Figure 2 do vary.
> > > >
> > > > - In Fig.2, we show the sampling results of two different distributions, therefore the scores are different;
> > > > - Fig.5 and Fig.6 correspond to the two experiments displayed in Fig. 2. Furthermore, since the slices of the samples of Fig. 5 (Fig. 6) come from the sampling of the same 3D density function, thus the scores, which represent the sampling quality, are the same across different rows of Fig. 5 (Fig. 6).

---

### Author Response · Authors · 2022-11-19
**The revised manuscript**

We have updated the manuscript based on the reviewers' comments and suggestions. The revised parts are marked as blue font in both the main paper and the appendix.

---

### Decision · Program_Chairs · 2023-01-20

**Decision:**

Accept: poster

**Justification For Why Not Higher Score:**

The application of OT in 3D are important in medical imaging domain and perhaps in robotics,  the applications of such technical work are not of general interest.

**Justification For Why Not Lower Score:**

N/A

**Metareview: Summary, Strengths And Weaknesses:**

Summary:

This  paper proposes an algorithm for numerically estimating the optimal transport (OT) map in dimension 3. In this context, the optimal transport map can be characterized as the solution of the Monge-Ampère equation. After discretization and linearization the paper builds on the work of  Loeper-Rapetti and Saumier et al to solve it using the FFT. A GPU implementation ensures a fast and parralelized implementation. The algorithm has a linear rate of convergence. Applications to density sampling and volume magnification are also provided in the paper.

Reviewers agreed on the merits of the paper. The paper improved a lot during the rebuttal thanks to the detailed reviews of reviewers Reviewer Q68w and Reviewer AQZc.  Thanks to them ! Many corrections to the rigor and the experimental setup made the paper in a better shape for publication.


**Note From Pc:**

if the above contains the word "oral" or "spotlight" please see: "oral" presentation means -> notable-top-5% and "spotlight" means -> notable-top-25%. As stated in our emails, we are disassociating presentation type from AC recommendations

**Summary Of Ac-Reviewer Meeting:**

N/A